# Fire risk modeling: an integrated and data-driven approach applied to Sicily.

Alba Marquez Torres[1,2], Giovanni Signorello[3,4], Sudeshna Kumar[1], Greta Adamo[1], Ferdinando Villa[1,5], Stefano Balbi[1,5]

[1]Basque Centre for Climate Change (BC3), 48940, Leioa, Spain.
[2]Department of Agriculture and Forest Engineering, University of Lleida, 25003, Lleida, Spain.
[3]University of Catania, Department of Agriculture, Food and Environment, Catania, 95131, Italy.
[4]Centre for the Conservation and Management of Nature and Agroecosystems (CUTGANA), Catania, 95123, Italy.
[5]Ikerbasque - Basque Foundation for Science, Bilbao, 48009, Spain.

*Correspondence to*: Alba Marquez Torres (alba.marquez@bc3research.org)

**Abstract.** Wildfires are key to landscape transformation and vegetation succession, but also to socio-ecological values loss. Fire risk mapping can help to manage the most vulnerable and relevant ecosystems impacted by wildfires. However, few studies provide accessible daily dynamic results at different spatio-temporal scales. We develop a fire risk model for Sicily (Italy), an iconic case of the Mediterranean basin, integrating a fire hazard model with an exposure and vulnerability analysis under present and future conditions. The integrated model is data-driven but can run dynamically at a daily time-step, providing spatially and temporally explicit results through the k.LAB software. K.LAB provides an environment for input data integration, combining methods and data such as Geographic Information System, Remote Sensing and Bayesian Network algorithms. All data and models are semantically annotated, open and downloadable in agreement with the FAIR principles (Findable, Accessible, Interoperable and Reusable). The fire risk analysis reveals that 45% of vulnerable areas of Sicily are at high probability of fire occurrence in 2050. The risk model outputs also include qualitative risk indexes, which can make the results more understandable for non-technical stakeholders. We argue that this approach is well suited to aid in landscape and fire risk management, both under current and climate change conditions.

## 1 Introduction

Fire, as a natural disturbance, has played an important role in shaping forest structure, increasing biodiversity and leading the species' evolution (Bond and Keeley, 2005; Pausas et al., 2004; Kelly and Brotons, 2017). However, the balance between the natural fire regime and the ecosystem is often disrupted when humans modify the environment to their needs. In recent years, the rural depopulation and simultaneous spread of urban areas as residential buildings into the countryside have increased the fire frequency and burned areas (Faivre et al., 2014; Robinne et al., 2016). Although this is a worldwide problem, the Mediterranean climatic area has experienced a great impact (Kocher and Butsic, 2017; Leone et al., 2009; Pausas and Fernández-Muñoz, 2012).

Sicily (Italy), the largest island of the Mediterranean Sea with 25,711 km², has been the cradle of several civilizations and their traditions, with continuous and intense human exploitation of natural resources (forestry, grazing, agriculture) (Antrop, 2005; Sereni, 1961), encompassing multiple agricultural and agroforestry landscapes (Baiamonte et al., 2015; Di Maida, 2020). Due

to its great variability of topography, lithology, pedology (Catalano et al., 1996) and climate (Bazan et al., 2015), Sicily is rich in biodiversity and ecosystems (Cullotta and Marchetti, 2007; Peruzzi et al., 2014). Therefore, the island can be viewed as representative of the Mediterranean basin as a whole.

Moreover, Sicily is the most populated island in the Mediterranean Sea with nearly 5 million inhabitants, similar to Denmark or Finland (Planistat Europe and Bradley Dunbar Association, 2003). As a consequence, year after year the environment has undergone degradation due to the increase of intensive farming practices, the urbanization growth in the most populated and tourist areas and the loss of traditional agricultural and forest management because of the rural population abandonment (Bazan et al., 2019; Falcucci et al., 2007; Prestia and Scavone, 2018). In the last 50 years, the increase of forest and scrub mass due to the abandonment of traditional land management (Bonanno, 2013; Ragusa and Rapicavoli, 2017) and the increase in the frequency of long droughts created optimal conditions for the occurrence of wildfires (Mouillot et al., 2005; Ruffault et al., 2020). The population living in the wildland-urban interface zone is particularly at risk due to exposure to fire and difficulty in evacuation.

Uncontrolled wildfires in Sicily have increased in recent years, making Sicily the Italian region with the highest number of fire events and the largest burned area between 2009 and May 2016 (Fig. 1). The probability of fire occurrence is mainly linked to ignition source, forest fuels and environmental conditions (Ganteaume et al., 2013; Hantson et al., 2015; Ricotta and Di Vito, 2014). The ignition sources are usually divided into natural causes (mainly lightning but geological causes too) and human (accidentally or intentionally) (Aldersley et al., 2011; Ganteaume et al., 2013; Rodrigues and de la Riva, 2014). The main causes of wildfires in Sicily are human-driven (Corrao, 1992; Ferrara et al., 2019). Arson and accidental wildfires, set up to create new pasture resources or to burn stubble, are the first causes of wildfires, especially in areas where vegetation interfaces with urban structures.

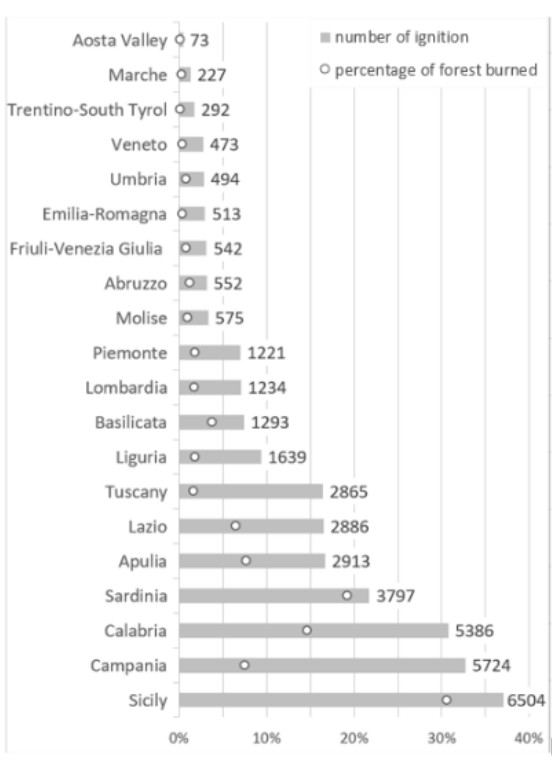

53

**Figure 1: Total number of fire ignitions and percentage of area burned (over 30 ha) in Italy by region between 2009 and May 2016. Source: Fire activity statistics, *Servizi AntiIncendio Boschivo* (Italian Fire Services).**

The consequences of wildfires exceed the loss of forest cover, vary over time and can be long-lasting. Some ecosystem properties and functions that deliver benefits to humans (Daily et al., 1997; Roces-Díaz et al., 2022), including biodiversity, may be lost. This diminish might happen when natural fire regimes and forest ecosystems are strongly altered by human intervention (Tedim et al., 2020; Arno and Brown, 1991), leading to an increase of fire extent, intensity and severity (Pausas et al., 2008; Regos et al., 2014; Castellnou et al., 2019). For example, after the wildfires in summer, with the arrival of the first heavy rains, there can be extensive erosion in burned areas, loss of organic matter or pollution of adjacent water bodies (Bisson et al., 2005; Certini, 2005). In general, burned areas lose their carbon sequestration capacity and desirability for outdoor recreation (Moreira and Russo, 2007).

The literature on fire modeling at different spatio-temporal scales is vast (Ganteaume et al., 2013; Jain et al., 2020; Tymstra et al., 2020). Due to its drought sensitivity, most studies focus on the Mediterranean climatic region (Oliveira et al., 2012; Satir et al., 2016; Wittenberg and Malkinson, 2009). Among the different methods applied, machine learning models are gaining traction due to increased computing power and data access. Many algorithms have been tested, including artificial neural networks, support vector machines, maximum entropy and random forest (Jain et al., 2020).

Risk fire mapping has been one of the most widely studied approaches in the forest fire literature. Even so, many models have become obsolete and have not been renewed (Ager and Finney, 2010; Mohajane et al., 2021). The spatial-temporal resolution is too coarse (Lozano et al., 2017) or does not take into account the distribution of forest fuel types (Bacciu et al., 2021; Michael et al., 2021), which is essential for risk reduction (Castellnou et al., 2019). Moreover, risk conditions for society are induced by progressive changes in environmental conditions. For this reason, it is indispensable to create open models that can incorporate new transdisciplinary data and knowledge (Nikolakis and Roberts, 2022; Wunder et al., 2021) that have arisen since 2016 (Artés et al., 2019; Duane et al., 2021).

On the society side, knowledge plays a key role in risk reduction, decision-making, coordinated policy action, and re-learning on fire. Vulnerability is associated with a lack of risk communication, especially a lack of sufficient information that can lead to a misunderstanding of risk (Birkmann et al., 2010). This has important implications for motivation and perceptual capacity to act and adapt to climate change (Grothmann and Patt, 2005). Moreover, understanding the fire risk processes can help society to comprehend the landscape transformation needed for a lower-risk environment (Otero et al., 2018). Although efforts are being made, few resources are allocated to the accessibility, sharing, and integration of knowledge at multiple scales across different stakeholders (Weichselgartner and Pigeon, 2015). Therefore, it is crucial to develop accessible tools and methods for fire risk assessment, where managers and stakeholders can consider social and environmental consequences.

Similarly, on the scientific side, lack of transparency has been one of the traditional characteristics of modeling (i.e. black box model), even within the decision support system leading to several scientific, organizational and ethical issues (Guidotti et al., 2018). Moreover, most of the models and resources developed by scientific research are not transferable or shared between different programming languages or modeling infrastructures. To connect the scientific knowledge, we applied the Integrated Modeling approach of ARtificial Intelligence for Environment & Sustainability (ARIES, https://aries.integratedmodelling.org/), which integrates a network of web accessible data, models, and other resources, implementing the FAIR principles (Wilkinson et al., 2016) through the k.LAB software, a semantic web-based modeling platform. The FAIR principles apply to the generated data and models, which must be:

- Findable: simple to identify by humans and computers;
- Accessible: easy access to metadata and resources stored;
- Interoperable: should be ready to be exchanged, interpreted and combined in a (semi)automated way with other datasets;
- Reusable: sufficiently well-described to be reused in future research and integrated with other data sources.

This study analyses wildfire activity for the years 2007-2020, to model fire risk in Sicily. We have adopted the definition of fire risk provided by the AR6 report of IPCC, i.e. the dynamic interaction between the components of 'climate related hazards with the exposure and vulnerability of the affected human or ecological system to the hazards' (IPCC, 2012). Thus, in this

article, we focus on answering three questions: where it is likely to occur?, what ecosystem services might be affected? and, what is the impact on the environment and the society?.

To this end, we have developed a set of models in the k.LAB software and we integrated them into the ARIES network. These models are modular, interconnected, and semantically explicit under k.LAB, where we simulated the current wildfires and their interaction with key human and biophysical drivers, using a machine learning algorithm. Furthermore, as proof of the advantages of using FAIR data and resources, it has been possible to analyze future fire risk under climate change and consider the consequences for different ecosystem services using models included in ARIES and developed by other experts.

## 2 Material and Methods

### 2.1 Study area

The case study was carried out on the island of Sicily, the largest and most populated island in the Mediterranean. Within its 2,571 ha, the altitudinal range reaches 3,357 m at the peak of one of the most active volcanoes in the world (Thomaidis et al., 2021). The island has a Mediterranean climate with mild and wet winters and dry and hot summers, highlighting the southwest coast, where the climate is affected by the African currents and summers. Rainfall is scarce leading to water deficits in some provinces. Moreover, the change in land use has gradually modified the climate, with less rainfall and drier rivers (Drago, 2005; Ragusa and Rapicavoli, 2017).

The land use change caused mainly by the intense deforestation throughout Sicily's history had favored intense agricultural practices, especially in the center and southwest. Thus, agricultural areas cover 57% of the island, whose 35% are arable lands and 22% permanent crops. Roughly a third of Sicily is forest, shrublands and open areas. Woodlands and semi-natural areas are sparse in the agricultural area and denser in areas with special protection, the most important being the Mount Etna surroundings, in the Nebrodi Mountains Regional Park and the *Natural Reserve of Bosco della Ficuzza* (Sicilia Assessorato beni culturali ed ambientali e pubblica istruzione, 1996). Due to its long-lasting socio-ecological history, location in the Mediterranean Sea, its fragility to climate change, and increasing fire regime, Sicily represents an ideal study area representative of the Mediterranean socio-ecological context.

### 2.2 Fire risk analysis

The interaction of environmental and social processes drives the risk (Table 1), determined by the combination of a physical hazard and the vulnerability of the socio-ecological elements exposed (IPCC, 2012).

**Table 1.  Fire risk is defined by vulnerability and hazard components (IPCC, 2012).**

| RISK |
|---|
| *The potential likelihood of negative consequences for the elements of value in a context considering the probability of occurrence of fire hazards. Fire risk results from the interaction of vulnerability, exposure, and* |

| | 1. Hazard | 2.Exposure | 3.Vulnerability |
|---|---|---|---|
| | | *hazard.* | |
| | *Probability of occurrence of a physical event (natural or human-induced) that may damage the elements in the same time-space context. For instance, the probability of fire occurrence.* | *Elements (and their values) that are in a context where a hazardous event, such as fire, may happen.* | *The tendency of exposed elements to be adversely affected by a hazardous event. For example, predisposition, susceptibility, fragility, weakness, of the exposed elements.* |
| | Fire hazard components:<br>• Weather:<br>  o Temperature<br>  o Weekly Maximum temperature<br>  o Days without precipitation<br>  o Weekly precipitation<br>  o Solar radiation<br>• Biophysical drivers:<br>  o Forest fuel<br>  o Elevation<br>  o Slope<br>• Human drivers:<br>  o Distance to protected area<br>  o Distance to road<br>  o Distance to human settlement | Ecosystem services exposed:<br>• Vegetation carbon mass<br>• Pollination<br>• Outdoor recreation<br>• Soil retention<br>• Biodiversity*<br><br>*Technically not an ecosystem service but added here as an associated element of exposure. | Vulnerability<br>• Wildland-Urban Interface (WUI)<br>• Wildland-Agricultural Interface (WAI)<br>• Nationally designated areas (CDDA) |

Fire hazard captures the probability of fire occurrence, based on historical wildfires and drivers such as biophysical factors
and human-modified areas. The fire hazard interacts with the elements exposed; we highlight exposed ecological values and
ecosystem services such as biodiversity, pollination, carbon mass, soil retention and outdoor recreation that may be affected
by fire occurrence.
Vulnerability identifies exposed elements that are more susceptible to being highly or irreparably damaged due to their intrinsic
or contextual characteristics. Wildland-Urban Interface (WUI) is particularly fire-prone because it is a forested area less than
200 meters from an urban area (Ganteaume et al., 2021; Intini et al., 2020), due to the relationship between the ignition points
and populated areas (Chappaz and Ganteaume, 2022). It also represents a high weakness for human settlement, as they are
extremely close to the forest, becoming a problem in fire management (Cohen, 2008). Wildland-Agricultural Interface (WAI)
is a forest area in close proximity (less than 200 meters) to an agricultural area and highly predisposed to burning due to the
fire used for clearing forest and pasture or crop establishment (Leone et al., 2009; Ortega et al., 2012). Moreover, fire impacts

agricultural land, making food safety susceptible to hazards (Baas et al., 2018). Natural areas with special protection (UNEP-WCMC and IUCN, 2022) are particularly fragile with species with different endemism ranges and sensitive to social, climate and environmental changes (Baiamonte et al., 2015).

Fire risk is considered to be the cumulative consequence given by the interplay context-specific elements. Those elements capture vulnerability, exposure, and hazard components emerging from the probability of fire occurrence. In this study we quantify fire risk by measuring the potential area affected, the hot spots of biodiversity and ecosystem services potentially exposed and their vulnerability. We also assess fire risk both in current and future conditions (S1, Fig. S1) to consider the impact of climate change.

**2.2.1 Fire hazard model**

The model presented in this study is developed using the k.LAB software to achieve interoperability from the data sources to the generated modeling results (Villa et al., 2017). Within k.LAB, an ontology-driven language called Knowledge-Integrated Modeling (k.IM), provides the basis for the semantic annotations (i.e., explicit definitions) of resources, such as external datasets, and individual modeling tasks (S1, Fig. S2). Once the resources are assembled in the resulting computational workflow, k.LAB returns in output contextualized models' results visualized on a map. To ensure transparency, textual documentation of the process followed to achieve the results with annexed references and the details about the workflow are also provided to the users.

Accurate spatio-temporal detection of fire hazard is essential for the modeling and analysis of fire risk; thus, a system that transparently keeps track of the origin and reliability of input data is crucial. The input data used in this study were collected from different sources and can be classified into two categories: (i) historical wildfires and (ii) explanatory variables, which include weather, human and biophysical drivers. The data collection and processing are discussed in the following paragraphs. All the data and resources are semantically annotated, openly accessible and interoperable within k.LAB.

Historical fire data from 2007 to 2020 were collected from two different sources: The Regional Agency of Fire Control in Sicily was used to identify the fire perimeter and the Fire Information for Resources Management System (FIRMS) satellite data to locate the ignition point (Table 2).

**Table 2. Information about historical fire data**

|  | Historical fire perimeter | Historical fire ignition |
|---|---|---|
| **Source** | Regional Agency of Fire Control in Sicily | FIRMS |
| **Spatial resolution** | GPS error, less than 10m | MODIS: 1km<br>VIIRS: 375m . |

| Temporal coverage and time consistency | 1 January 2007 - 31 December 2020 (Daily) | MODIS Collection 6: 11 November 2000 – present (Daily)  VIIRS: 20 January 2012 – present (Daily) |
|---|---|---|
| Coordinate Reference System (CRS) | EPSG:102092 - Monte_Mario_Italy_2 - Projected for the years 2009 and 2017: EPSG:3004 - Monte Mario / Italy zone 2 - Projected | EPSG:4326 - WGS 84 - Geographic |
| Feature Type | Polygon | Point |

The regional agency collects the perimeter data of historical fires and provides the fire start and end dates collected by the
Forestry Information System (SIF – *Sistema Informativo Forestale*) and the forestry command corps of the Sicilian region
(*Comando Del Corpo Forestale Della Regione Siciliana*). FIRMS was developed by the University of Maryland, to locate
active fires in near real-time by data from MODIS (Moderate Resolution Imaging Spectroradiometer) and VIIRS (Visible
Infrared Imaging Radiometer Suite) (Giglio et al., 2016; Schroeder et al., 2014). MODIS is an instrument aboard Terra and
Aqua satellites that provides global coverage every 1-2 days and VIIRS sensor is on board the Suomi JPSS-1 satellites and
provides full global coverage every 12 hours. When there was information from both satellites for the same fire perimeter,
VIIRS was prioritized. Due to its spectral and spatial resolution, VIIRS sensor is more accurate in fire detection (omission and
commission of errors) thanks to the detection of the radiative power of the fire, especially in low biomass areas (Fu et al.,
173  2020).

Satellite data was used to locate the fire ignition point inside the perimeter provided by the regional agency. The centroid was
considered the ignition point for the perimeters when it wasn't identifiable using satellite data. To prevent double-counting
from the data sources, each fire perimeter was double-checked to verify that there was only one ignition point fire perimeter.
We obtained a total of 7,492 points linked with their ignition date (day, month and year).
In addition to the ignition data, we prepared an equal number of locations without fire events. This is needed to preserve a
balanced dataset of observations that considers the explanatory variables values both in case of ignition and the absence of
ignition. The result of an imbalanced training dataset is a "skewed data bias" and a model not capable of discriminating relevant
patterns in data (Rennie et al., 2003). The weights for the class with less training data, will be lower when the training data is
skewed. Consequently, classification will be unfairly biased in favor of one class over another. The learning algorithm becomes
too specific, leading to overfitting (Li et al., 2021).
The points without ignition were randomly generated with seeds within the study area between the 01-01-2007 and 31-12-
2020 periods. It was verified that none of these points overlap with historically burned perimeters in date and location. The
"ign" attribute differentiates ignition points (1) from non-ignition points (0) (Fig. 2).

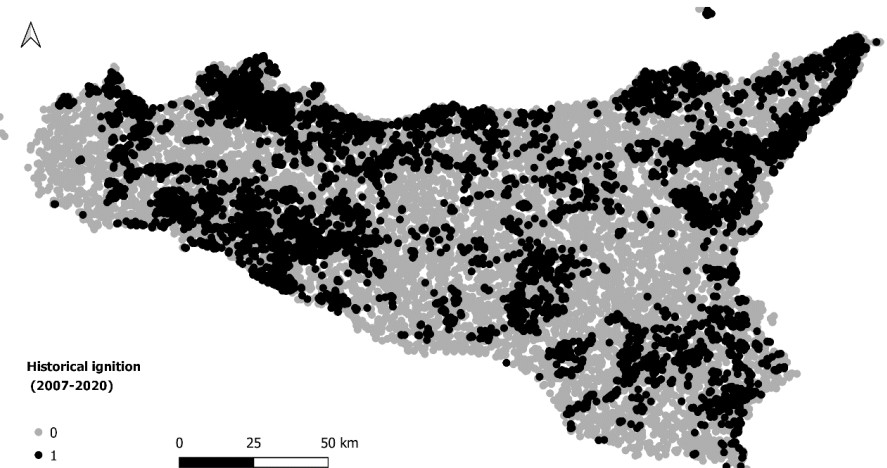


**Figure 2: Distribution of historical wildfires (category 1, black color) and no wildfires (category 0, grey color) in the Sicily region**
**from 2007 to 2020.**
The data feeding the machine learning model comes from open resources on the cloud provided by well-known and reliable
institutions. Those input data are incorporated automatically, depending on the spatio-temporal needs of the model. In the
Sicily model, the data comes from the Regional Government of Sicily, the University of Catania or E-OBS (Ensembled
OBservation) project, among others (Table 3).
**Table 3. Variables in the BN model**

| Variable (semantic language) | Description | Type | Unit | Source |
|---|---|---|---|---|
| occurrence of Fire within Site | Present and absent | Discrete | 1 (fire) - 0 (no fire) | ARIES[a], SIF[b] and FIRMS[c] |
| Atmospheric Temperature | Mean temperature | Continuous | Celsius degrees | E-OBS[d] |
| Weekly Maximum Atmospheric Temperature | Mean of maximum temperature in the last week | Continuous | Celsius degrees | ARIES[a] (based on E-OBS[d] data) |
| count of Day | Counting days | Continuous | # | ARIES[a] (based on |

| | | | | |
|---|---|---|---|---|
| without Precipitation | since last precipitation | | | E-OBS[d] data) |
| Weekly Precipitation Volume | Accumulated precipitation during a week | Continuous | mm | ARIES[a] (based on E-OBS[d] data) |
| Solar Radiation | Total solar radiation | Continuous | J/m^2 | E-OBS[d] |
| value of Forest during Fires | Forest fuel type | Discrete | see in S2, Table S1 | University of Catania |
| Elevation | Geographical elevation above sea level, as described by a digital elevation model | Continuous | m | SITR[e] |
| Slope | Inclination of the above-water terrain in a geographical region | Continuous | grade | ARIES[a] (based on elevation from SITR[e]) |
| distance to ProtectedArea | Distance to protected area | Continuous | m | k.LAB[f] (based on OSM[g]) |
| distance to Road | Distance to road | Continuous | m | k.LAB[f] (based on OSM[g]) |
| distance to Human Settlement | Distance to human settlement | Continuous | m | k.LAB[f] (based on OSM[g]) |

[a] ARIES: ARtificial Intelligence for Environment & Sustainability
[b] SIF: *Sistema Informativo Forestale* (Forestry Information System)
[c] FIRMS: Fire Information for Resources Management System
[d] E-OBS: Ensembled OBservation
[e] SITR: *Sistema Informativo Territoriale Regionale* (Regional Spatial Information System)
[f] k.LAB: Knowledge Laboratory
[g] OSM: Open Street Map (OpenStreetMap contributors, 2020)
In the case of Sicilian wildfires, the human factor is one of the main triggers that lead to the depopulation of country areas by
land managers and the increasing number of tourists and visitors. The human drivers used as explanatory variables in the model
are distance to protected areas, distance to road and distance to human settlement. Those variables are calculated using
semantics in the k.LAB software. K.LAB is able to compute geographical distances (Euclidean distance) between spatial

objects. Additionally to human drivers, the fire hazard also depends on (i) weather (especially due to long dry seasons), (ii) topography and (iii) environment, characterized by the high flammability of the Mediterranean forests (Corrao, 1992). Some of the weather variables, based on E-OBS data, were integrated into the ARIES network. Those drivers influence fuel type, moisture levels and fire behavior.

Meteorological data were obtained from the E-OBS Copernicus project (Cornes et al., 2018). We used the last version released in March 2021 to obtain data from 2007-01-01 to 2020-12-31. The data were processed with R software to obtain the meteorological data needed on each specific day as daily temperature and daily solar radiation (Table 3).

In addition, heatwaves and long periods of drought are great drivers for the majority of extreme wildfires (Narcizo et al., 2022; Nojarov and Nikolova, 2022; Parente et al., 2018). Moreover, with climate change, these episodes will increase in number, frequency and intensity, especially for the projections for RCP 8.5 (Molina et al., 2020). We have taken into account variables such as the mean of maximum temperatures, the number of days without rain and the precipitation accumulated during the previous week.

The topographic factors used (slope and elevation) are constant components of the fire risk model. They have a strong influence on other parameters such as fuel conditions and weather. Slope and elevation were generated from a Digital Elevation Model (DEM) at a 10 meters' resolution.

Fuel type and land cover composition have a significant effect on fire ignition. Deep knowledge of the fuel bed is key to fire management, as it is one of the main components of fire risk. Fuel bed has been reformulated into fuel models for easier use in models and systems. The characteristics and properties of fuel types used categorical ranges between 1 to 7 (S2, Table S1) according to the Prometheus project (Lasaponara et al., 2006). The latter defines fuel type as a recognizable combination of fuel components with distinct species, shapes, dimensions, structures, and continuity that will display a particular fire behavior under specific burning conditions (Merrill and Alexander, 1987). The land cover map source is based on the Italian Nature Map (Angelini et al., 2009). Landcover is mainly composed of extensive crops and complex farming systems (46%) so, the main fuel type is ground fuels such as grass (50% of land in Sicily). 29% of the land cover on the island is non-combustible.

Among the models that were tested, one of them had the fire frequency as input, calculated with the historical wildfires from 2007-2020. This model had an accuracy above 95%. After several literature searches and discussions with experts, it was decided not to incorporate fire frequency into the model. Although the accuracy was much better than the model finally chosen (83.6%), the main disadvantage was the possibility of overfitting. In addition, it may lower the likelihood of detecting wildfires in unusual areas due to changes in land use or phenomena such as climate change. Finally, the difficulty of accessing new wildfires to incorporate into the frequency variable was another important reason.

### 2.2.2 A Bayesian Network model of fire hazard

Bayesian Networks (BN) (Pearl, 1988) have been widely used in recent years and have been highlighted as a powerful tool for modeling complex problems, representing uncertainty and assisting stakeholders when the data is highly interlinked (Henriksen et al., 2007; Kangas and Kangas, 2004; Penman et al., 2011). Thus, the BN model is especially useful in environmental modeling as wildfire risk because (i) involves a high level of uncertainty, (ii) has limited or incomplete data on key system variables, (iii) contains both qualitative and quantitative information or data in different forms, and (vi) integrates multidisciplinary systems (Chen and Pollino, 2012). In addition, the system is transparent in its process, as its nodes table shows the dependency's strength between nodes and their parents in terms of conditional probability distribution and the relationships between variables are made explicit.

A BN is a model that graphically represents causal assertions between variables as patterns of probabilistic dependencies. The Directed Acyclic Graph (DAG) of a BN is built with nodes (variables) and edges between the nodes (dependencies and mutual relationships between variables). Each successor node (children) is only determined by the values of its immediate predecessors (parents) known as parental Markov property (Pearl, 2009). Roots are the nodes without any parent and with marginal distribution (Borsuk, 2008).

The BN has been learned using the WEKA (Waikato Environment for Knowledge Analysis) library integrated into the k.LAB software (Bouckaert, 2004; Frank et al., 2016; Willcock et al., 2018). WEKA is an open source JAVA library providing a collection of machine learning algorithms. The WEKA interface provides graphical and text components to inspect some BN's properties as basic algorithm information, the BN structure, the probability distribution table or the accuracy by class.

The model has been written in a semantically explicit way using the aforementioned k.IM language (S1, Fig. S2), which compiles in Web Ontology Language (OWL) (Bao et al., 2012) and allows to ontologically define and model natural language-like logical expressions. In addition, a model written in k.IM is able to interoperate with other models available in the k.LAB environment. When modeling in k.IM concepts that have been previously defined in a knowledge-base are invoked, examples are *earth:Site* and *chemistry:Fire* as depicted in (S1, Fig. S2). Those concepts carry out meanings facilitating a semantic integration within the system (Villa et al., 2017).

Since the BN is built with categorical values, continuous data need to be discretized. Discretization allows the establishment of non-linear values between variables and more complex distributions (Friedman and Goldszmidt, 1996). Discretizing the data helps to interpret the results more easily when it comes to decision-making processes by facilitating communication between modelers and end users. However, the interval selection interferes with the final results. We have been taking into account that the higher the number of intervals, the more data is needed to find significant dependencies (Aguilera et al., 2011); the nodes become weak when there are many intervals because there is less data for each distribution.

Among the methods to discretize (Beuzen et al., 2018), in this study we use both the equal-width and equal-frequency binning unsupervised methods, according to the input data distribution (see the data histograms in S3). In the first case, the algorithm divides the data into k intervals of equal size and in the case of equal frequency, the user specifies the sub-ranges that result in k intervals (bins) with approximately the same number of values. After modeling with different discretization ranges and obtaining similar accuracy results, we have chosen for each of the variables the minimum number of intervals in order to keep ecological sense, statistical significance and minimize information loss (S4, Table S2). The discretization applied is shown in

**Table 4. Discretization applied to the variables used in the fire occurrence modeling.**

| Semantic | Method | Bins |
|---|---|---|
| AtmosphericTemperature in Celsius | equal-width | 10 |
| Weekly Maximum AtmosphericTemperature in Celsius | | 10 |
| SolarRadiation in J/m^2 | | 5 |
| Weekly PrecipitationVolume in mm | | 10 |
| Count of Day without Precipitation | equal-frequency | 5 |
| Slope in grade | | 5 |
| Elevation in m | | 5 |
| distance to ProtectedArea in m | | 5 |
| distance to Road in m | | 5 |
| distance to Human Settlement in m | | 5 |

To learn the BN, 80% of the dataset was used to actually learn the model and 20% to test the relationship between historical wildfires (observations) and explanatory variables. On the learning side, we selected the K2 algorithm (Cooper and Herskovits, 1992). This type of score-based algorithm searches for the most probable belief-network structure through a heuristic search. The K2 algorithm processes each node in turn and greedily considers adding edges from previously processed nodes to the current one, adding the edges that maximizes the network's score. It turns to the next node when any of the following

requirements are met: (i) it has reached the maximum number of parents, (ii) there are no more parents to add, (iii) the score
has not improved (Chen et al., 2008). The number of parents for each node can be restricted to a predefined maximum (e.g.
maxparents = 1) to mitigate overfitting.
The BN predictors have been distributed in a Directed Acyclic Graph (DAG) as shown in Figure 3. DAG is assigning
probabilities to each variable's predictor; anthropogenic and biophysical factors such as meteorology, topography and
environment. The most influential variable of a BN results from the following characteristics: (i) the strength of influence of
each edge connecting the nodes (Balbi et al., 2019) and (ii) how "far", in terms of number of edges, is an input node from the
final output (Marcot et al., 2006). The strength of influence is calculated from the conditional probability tables and expresses
the difference between the probability distributions of two nodes by looking at the posterior probability distribution of a node,
for each possible state of the parent or child node. To summarize this difference, we report normalized Euclidean distance,
although other types of distances (e.g. Hellinger) are also used (Balbi et al., 2019). Table 5 quantifies numerically the strength
of influence as the thickness of the edges between Fire Hazard node and its children. The predictors with the highest strength
of influence are (i) atmospheric temperature, (ii) days without precipitation, (iii) fuel type and (iv) solar radiation (Table 5),
all of which are directly linked to the final output (fire occurrence). While atmospheric temperature, number of days without
precipitation, and solar radiation are expected to increase in variability and increase fire hazard with limited options for human
mitigation, fuel type can be managed with punctual landscape interventions reducing its combustibility level where it is more
necessary.

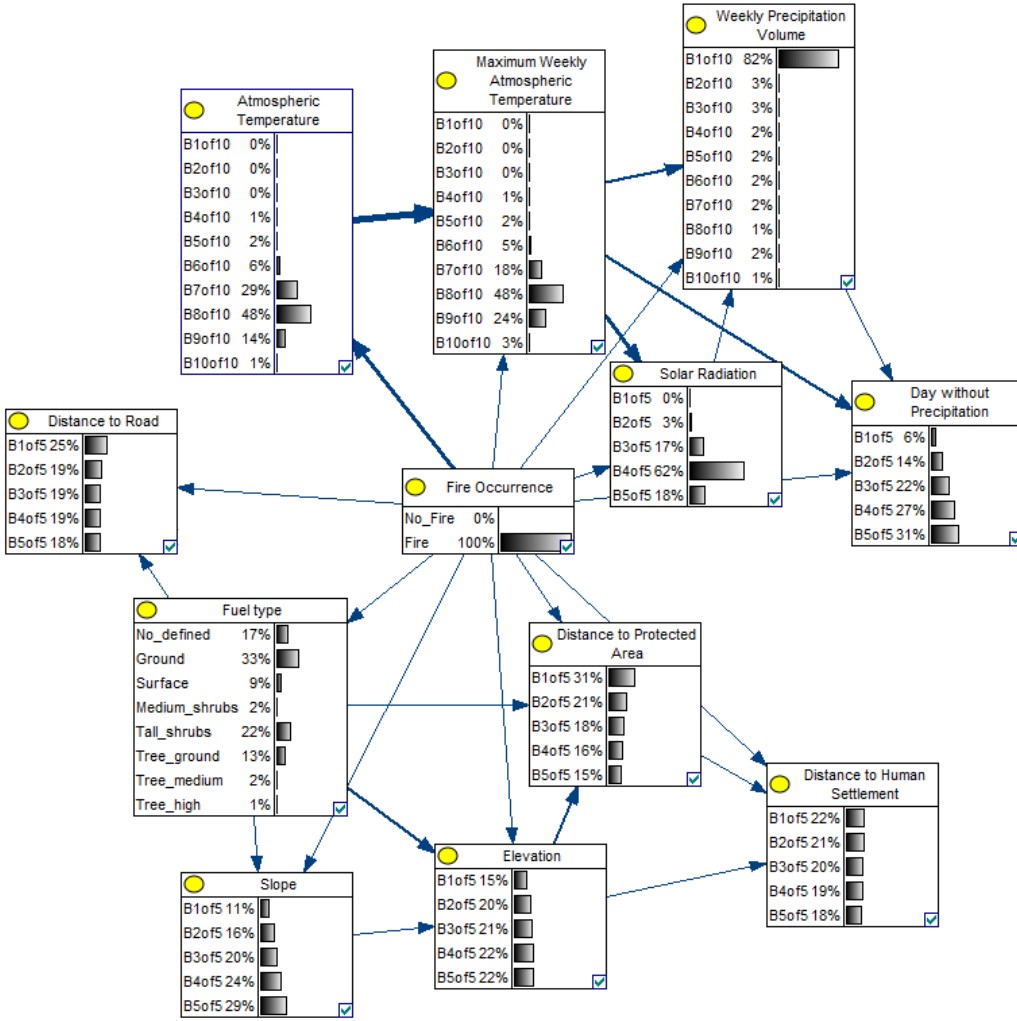


Figure 3. Directed Acyclic Graph (DAG) of the fire hazard Bayesian Network model where arcs width shows the strength of influence between nodes. Nodes show the relative probability of each interval of the variable, described in Supplementary Materials (S4, Table S2).

Table 5. Strength of influence between fire occurrence and its child nodes.

| Variable | Strength of influence |
|---|---|
| Atmospheric Temperature | 0.338 |
| Day without Precipitation | 0.193 |
| Fuel type | 0.192 |

| | |
|---|---|
| Solar Radiation | 0.191 |
| Elevation | 0.158 |
| Maximum Weekly Atmospheric Temperature | 0.154 |
| Distance to Protected Area | 0.145 |
| Slope | 0.138 |
| Distance to Road | 0.117 |
| Weekly Precipitation Volume | 0.113 |
| Distance to Human Settlement | 0.112 |

Finally, to be more understandable for end-users and stakeholders, the results of the model were divided into 3 equal intervals,
related to the level of fire occurrence (high: more than 66% chance, medium: between 33 and 66%, low: probability of fire
less than 33%).
**2.2.3. Drivers of vulnerability and exposed elements**
Social and environmental vulnerability have been assessed as the tendency of exposed elements to be potentially damaged by
a fire hazard due to its intrinsic or contextual conditions (IPCC, 2012). First, we used models developed in previous projects
in k.LAB to determine the socio-ecological exposed elements. The ecosystem services models and biodiversity considered are
those included in the ARIES global model set (Martínez-López et al., 2019). Once the fire hazard model is in k.LAB, all the
data and models can interoperate between them through the explicit semantics (Villa et al., 2017). Thus, we can reuse previous
ecosystem services models developed (Martínez-López et al., 2019; Willcock et al., 2018) applying them to a different context
and creating new knowledge. In this case, due to the specificities of Sicily and the relevance of ecosystem services affected by
wildfires, we choose to consider the following models: (i) vegetation carbon mass, (ii) pollination, (iii) outdoor recreation, (iv)
biodiversity and (v) soil retention. These models, published in (Martínez-López et al., 2019; Willcock et al., 2018), are briefly
described below:

• Vegetation carbon mass: calculates the above- and below-ground carbon storage in vegetation (T/ha), in accordance

with Tier 1 Intergovernmental Panel on Climate Change (IPCC) methodology (Gibbs and Ruesch, 2008; IPCC, 2006).

• Pollination: based on land use, cropland, and weather patterns, the pollination model generates spatially explicit data

of the supply and demand for insect pollination services.

• Outdoor recreation: calculates the accessibility of recreational features of the natural landscape, and the demand for

them, based on the methods by (Paracchini et al., 2014).

- Soil retention: the model provides biophysical estimates of soil loss and retention by plants (in tons of sediment per hectare per year) using the widely used Revised Universal Soil Loss Equation (RUSLE; (Renard et al., 1997).
- Biodiversity: a Bayesian Network approach used to learn from site-based expert estimations of "biodiversity value" to create a map of the entire Sicilian region (Willcock et al. 2018).

To create a comprehensive indicator of ecosystem services and biodiversity, we converted the above-mentioned modeling output to a common scale, using quantitative and qualitative criteria. In order to calculate the potentially reduced social and ecological services, we used the normalization method, instead of others such as qualitative categorization and probabilistic approaches (normal, Poisson, binary) (Chuvieco et al., 2003). We transformed each modeling output rescaling it from 0 to 1, using the minimum and maximum value within the Sicily context. The quantitative scale was classified into 3 categories (1-low, 2-medium, 3-high) using equidistant intervals; thus integrating all modeling outputs into a single value. In this quantitative cross-assessment, the most valuable component was prioritized. The final map was overlaid with wildland areas.

Once exposure was identified, we located the most vulnerable elements that were exposed to fire. Spatial data were generated for WUI, WAI and protected areas. In order to create the WUI area, we generated a 200 m buffer map from the human settlements, then overlaid it with the forest areas. The WAI map followed the same procedure, but with the buffer map from the agricultural areas. Finally, we use the FAO map (UNEP-WCMC and IUCN, 2022) for the protected areas. Vulnerable areas were overlapped with the exposure map.

Finally, the fire hazard model was used to predict how the most vulnerable exposed elements could be affected in the current and future climatic conditions. The future climate data was drawn from the Coupled Model Intercomparison Project 5 (CMIP5) for RCP 8.5 from COordinated Regional climate Downscaling EXperiment (CORDEX) (Giorgi et al., 2009). The data are bias-corrected and simulated by state-of-the-art global and regional climate model pairs. To generate the climatic variables, we used the same process as the current variables. We kept the other variables (solar radiation, fuel, slope, elevation, distance to road, protected area and human settlement) with the current conditions.

## 3. Results

### 3.1. Historical data analysis

During the analysis period (2007-2020) 28,814.698 ha were burnt in 12,749 fire perimeters and the data shows significant variability between years (Fig. 4). The average area burnt is equivalent to 20,630 ha with 910 ignitions per year, 2012 being the worst year, with 1,274 ignitions and 55,699 ha burnt. However, the monthly distribution over this period is skewed toward

July and August (Fig. 5), due to the weather's favorable fire conditions. August is clearly the month with more wildfires in all the years analyzed, with 4,166 ignitions and 118,481 ha burnt in total (26% more area than July, the second worst month).

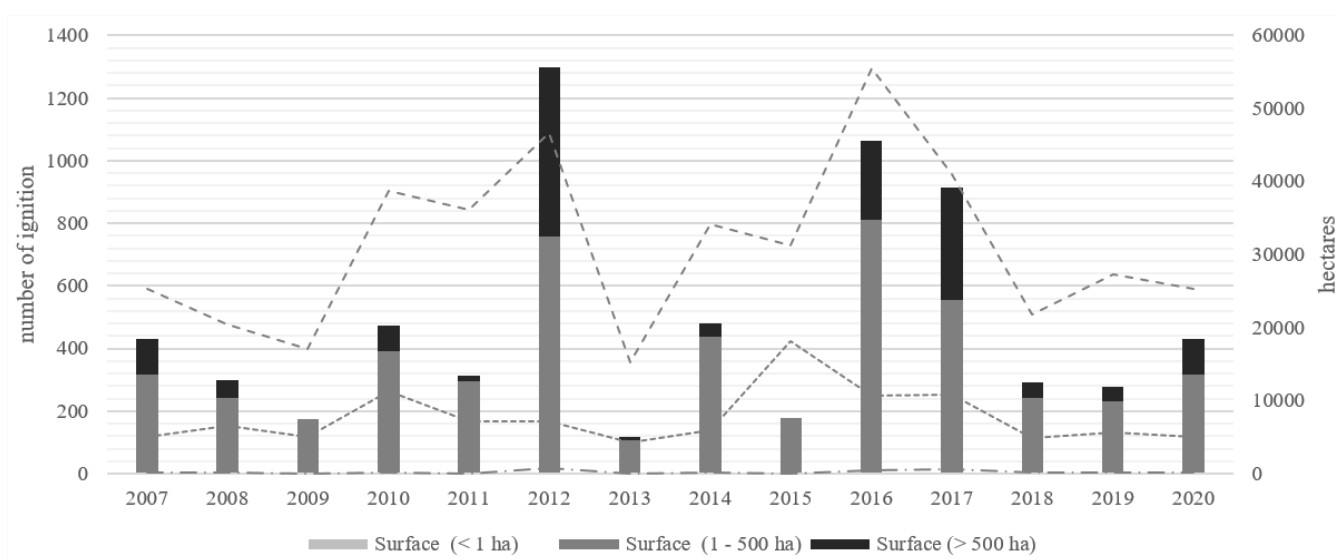

**Figure 4: Number of ignitions vs. burned area by year from 2007 to 2020. Source: Regional Agency of Fire Control in Sicily and FIRMS.**

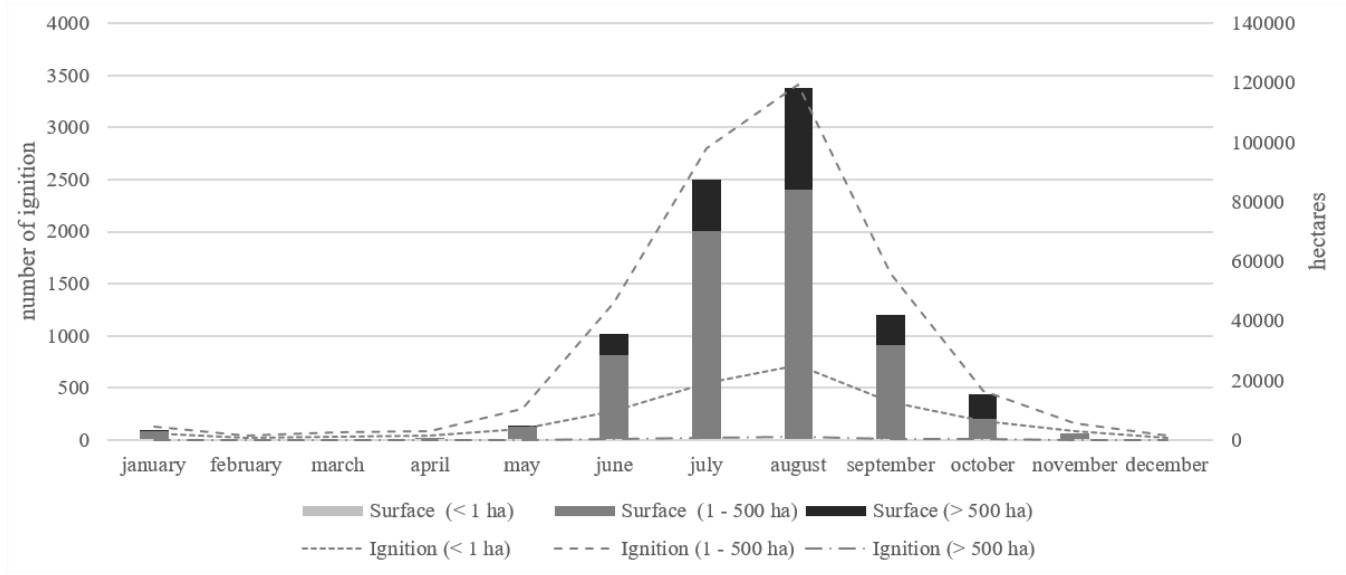

**Figure 5: Historical ignitions vs. burned area by month from 2007 to 2020. Source: Regional Agency of Fire Control in Sicily and FIRMS.**

Fire frequency analysis (Fig. 6) showed that a quarter of the area affected during 13 years (from 2007 to 2020) has burnt once,
34.8% twice. 23.1% have burned three times or more, and nearly 6% have been burnt more than 5 times in 13 years. Burned
area is spread throughout Sicily, however, areas close to cities, such as Palermo, have been burnt more than others.

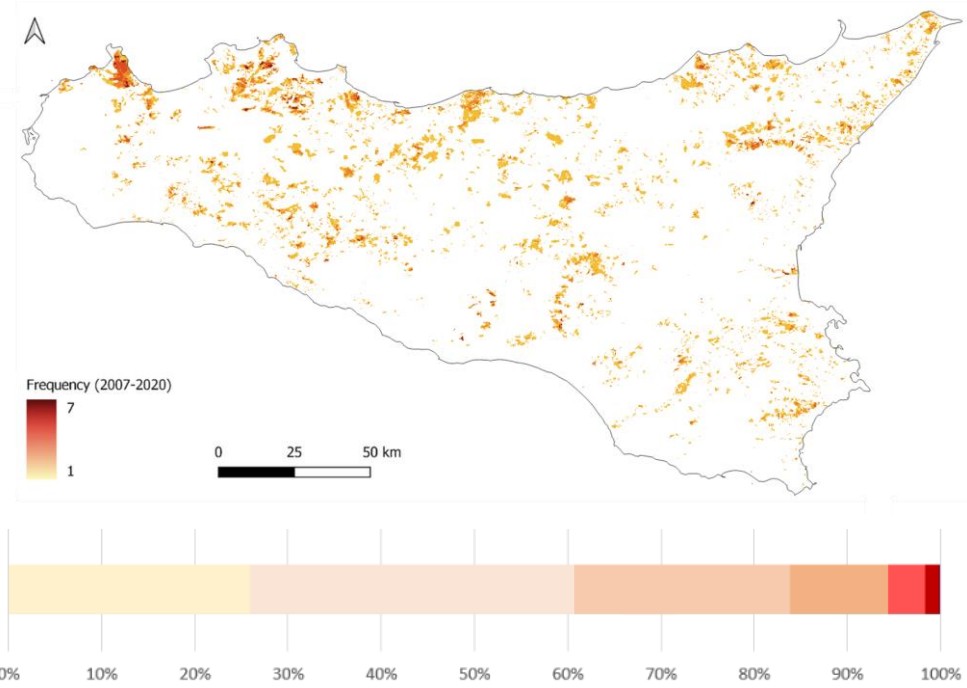



**Figure 6: Fire frequency aggregated by year. The legend shows how many times the same area has been burnt during the period of 2007-2020.**

Fire ignition causes have been recorded since 2010. Figure 7 shows that, every year, more than 70% of wildfires are caused
by arson, with 2010, 2011 and 2012 being particularly relevant. The percentage of wildfires caused by negligence or natural
effects is of little relevance. In general, it seems that the trend of arson is decreasing significantly over the years, from 91.54%
to 67.06%. A large part of the percentage that decreases due to arson is replaced by wildfires of unknown origin, so we cannot
be confident that this trend is real.

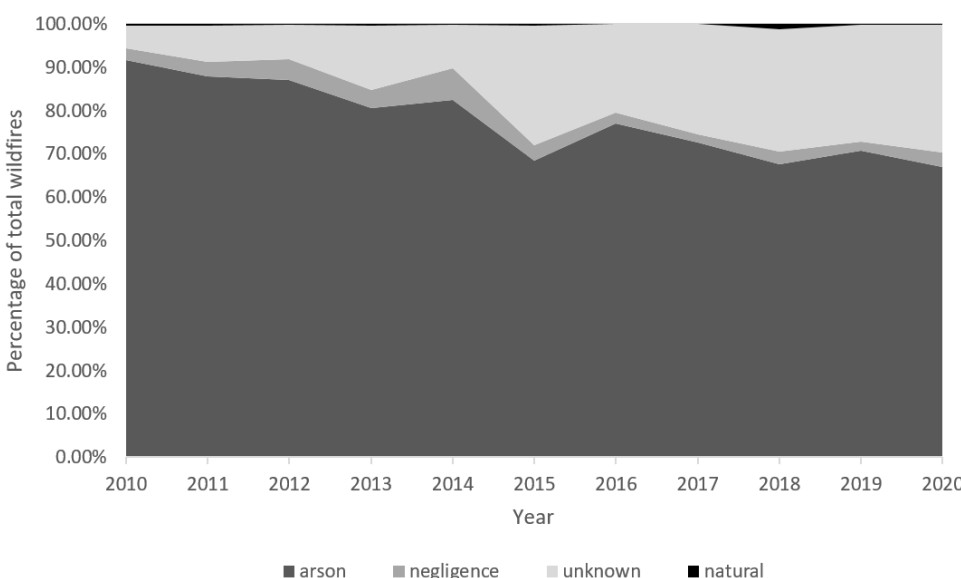


**Figure 7: Relative frequency of fires by main fire causes in Sicily in 2010–2020.10 the total wildfires**

### 3.2. The Bayesian data-driven approach

The Bayesian network model shows the probability of each child node under the probability of fire occurrence (where fire hazard is the parent node). For this purpose, in Figure 3, the state of the parent node based on historical fire is set to 100%, indicating that wildfire is certain. The posterior probability of each state of the explanatory variables is then obtained given the Conditional Probability Table (CPT) of each node (Fig. 3).

Accordingly, the fire occurrence probability at "Atmospheric temperature" is highest between 24.71ºC and 28.65ºC (S4, Table S2) and the weekly maximum temperature is between 27.93ºC and 31.69ºC. In over 80% of the cases, the weekly precipitation accumulated is below 0.05 mm for fire occurrence. Moreover, the more days without precipitation and higher solar radiation, the higher the probability of fire occurrence. As for the topographic variables, the most important is the slope, since the probability of fire is directly proportional to the slope. The same is observed in the case of elevation but in a less obvious pattern. The probability of fire is higher in locations that are closer to human activities such as roads or buildings and protected areas. Finally, in the case of the environmental variables, the highest fire probability in fuel forest type (S2, Table S1) is when ground fuel is grass (type 1), followed by high shrubs (between 2.0 and 4.0 m) and young trees resulting from natural regeneration or forestation (type 4). The third riskier fuel type is type 5, which occurs when the ground fuel is removed either by prescribed burning or by mechanical means. This situation may also occur in closed canopies in which the lack of sunlight inhibits the growth of surface vegetation.

The most influential variables (in terms of connection strength) according to our BN algorithm are atmospheric temperature, days without precipitation and fuel type (Table 5). While atmospheric temperature and the number of days without precipitation

are expected to increase in variability and increase fire hazard with limited options for human mitigation, fuel type can be
managed with punctual landscape interventions reducing its combustibility level where it is more necessary.
The k-fold cross-validation algorithm has been used to estimate the model's accuracy. This algorithm uses the training/testing
process "k" times and averages the results. The results for k=10 showed that 83.997% of the instances were correctly classified
in two values: occurrence and non-occurrence of wildfires.
We use the confusion matrix to measure the performance of the classification (Table 6). The results show 12,172 correctly
classified instances, but also 1,426 false positives and 893 false negatives. The type I error (false positive), i.e. detecting a fire
where it, in reality, is not, could lead to allocating efforts to unnecessary areas. Type II error (false negative), could not identify
the probability of fire in risk situations and, therefore, would not be managed properly. A false negative rate (0.11) is calculated
as the number of incorrect positive predictions divided by the total number of negatives; the best false positive rate is 0.0.
**Table 6: Confusion matrix of fire hazard BN modeling.**

| | | Real | | |
|---|---|---|---|---|
| | | **No fire** | **Fire** | *Sum* |
| **Predicted** | **No fire** | 5,573 | 893 (type II error) | *6,466* |
| | **Fire** | 1,426 (type I error) | 6,599 | *8,025* |
| | *Sum* | *6,999* | *7,492* | *14,491* |

The Bayes theorem is key to interpreting the output of binary classification problems using the calculated confusion matrix.
Precision is the confusion matrix probability P(Fire/TotalPredictedFire) = 6,599/8,025 = 0.822. It is the probability that the
fire predicted as fire is true. Recall P(Fire/TotalActualFire) = 6,599/7,492 = 0.881 is the percentage of the actual fires that were
correctly predicted by our classification algorithm. Table 7 also shows that the precision for the negative class (no fire) is
0.822. Moreover, the overall accuracy (weighted average between fire and no fire) is 0.841 and 0.840 for precision and recall
respectively and gives an overall picture of our model. These weighted results are close to our precision and recall values for
fire variables because our model is balanced (7,492 wildfires (51.70%) vs. 6,999 no wildfires (48.29%)). Hence, the overall
accuracy (0.84) is a good metric in this situation.
**Table 7: Sensitivity analysis of fire hazard model.**

| | **TP** | **FP** | **Precision** | **Recall** | **F-** | **MCC** | **ROC** | **PRC** |
|---|---|---|---|---|---|---|---|---|

| | Rate | Rate | | | Measure | | | |
|---|---|---|---|---|---|---|---|---|
| **No fire** | 0.796 | 0.119 | 0.862 | 0.796 | 0.828 | 0.681 | 0.915 | 0.922 |
| **Fire** | 0.881 | 0.204 | 0.822 | 0.881 | 0.851 | 0.681 | 0.915 | 0.903 |
| Weighted Avg. | 0.84 | 0.163 | 0.841 | 0.84 | 0.84 | 0.681 | 0.915 | 0.912 |

The confusion matrix is also useful for measuring other significant metrics such as the ROC (Receiver Operating
Characteristic) curve that summarizes the performance of the Bayesian classifier over all possible thresholds (Bradley, 1997;
Fawcett, 2006). It measures accuracy in a weighted sort and is appropriate when the observations are balanced between each
class, as in this case. For example, we used a sorting-based method called Area Under the ROC Curve (AUC) that measures
the two-dimensional region below the ROC curve from (0,0) to (1,1). Not only the model presents a strong AUC result of
0.915 for fire hazard, because the result is close to 1, but it also shows a significant F-Measure (a harmonic mean of the
precision and recall) with 0.847. The model performs well also in terms of uncertainty of the results. In Supplementary
Materials (S4, Figure S6) we display the uncertainty map associated with the standard deviation of the probability distribution
of fire hazard.
As an example, we present the fire hazard model results (i.e. the mean values of the simulated probability distributions) for
August 2050 because this is the month with the most critical historical wildfires in Sicily (Fig. 5), assuming no changes in
ecosystem management. Given the ease of access and reuse of models and data in k.LAB, any user of the modeling platform
can run the fire hazard model at any time in the future until 2055, as the input data are on the platform and are openly available.
As anticipated, the results of the model were divided into 3 equal intervals, related to the level of fire hazard (low: probability
of fire less than 33%, medium: between 33 and 66%, high: more than 66% of chance). Figure 9 shows the comparison between
the average results for August in 2020 and 2050 at 50 m of resolution.

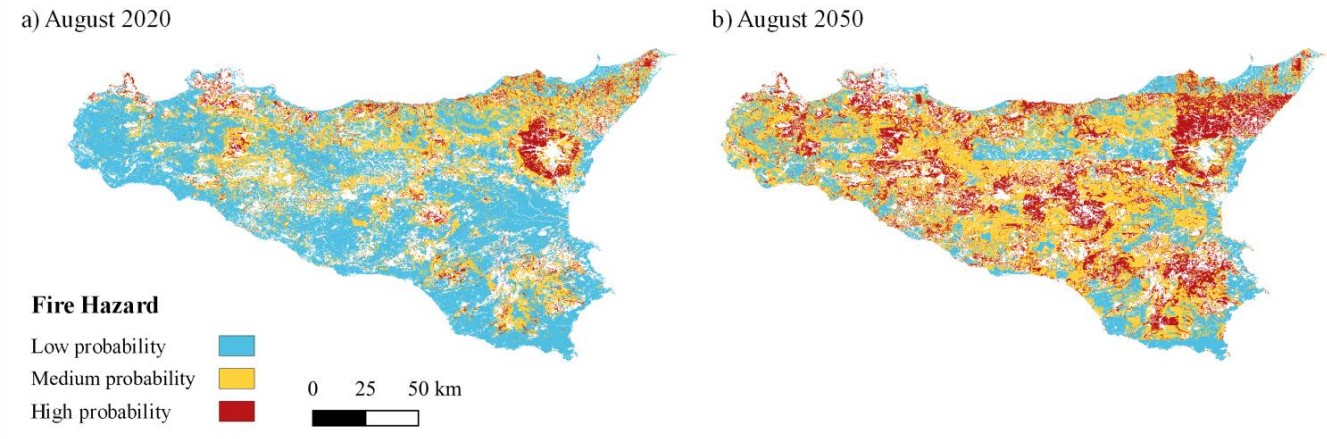

a) August 2020      b) August 2050

**Fire Hazard**

Low probability
Medium probability
High probability

0   25   50 km


**Figure 9. Example of fire hazard in (a) August 2020 (b) and 2050 classified by low, medium, or high probability of fire occurrence.**
When comparing simulated outcomes for 2020 and 2050, the increase of areas with high fire probability and decrease of those
with low fire probability becomes evident. The area with low fire probability changed from 12,300 km² to 4,887 km²,
representing a reduction of almost 40%. The extension of area with medium probability of fire occurrence increased from 29%
of the total wildland area to 48%. Finally, the wildland area with high fire probability occurrence changed from 8% (1,675.26
km²) to 27% (5,357.62 km²), an increase of 319.8% between the two scenarios. We here highlight the most significant change:
from low to medium probability of fire occurrence, which has increased by 7,112.58 km². Conversely 4,504.34 km² of wildland
areas with low probability of fire occurrence remain unchanged between 2020 and 2050.
**3.3. Wildfire risk levels**
The wildfire risk map at 50 m of resolution integrates a set of variables related to exposure and vulnerability (Table 1). In this
study, we analyze the areas with important ecological values and ecosystem services for both humans and nature, which would
be potentially affected in case of fire due to its exposition.
Figure 10 compares the average spatial variability of the ecosystem services and ecological values exposed in August 2020
and August 2050. In the horizontal axes, the figures are distributed by levels of fire occurrence probability. (low, medium, and
high), according to the fire hazard model. The 2020 column shows that the most exposed area corresponds to the low fire
hazard level. As the level of fire hazard increases, the exposed area decreases. In contrast, the 2050 column shows that the
most exposed area corresponds to the medium fire hazard level, followed by high and low probabilities of fire occurrence.

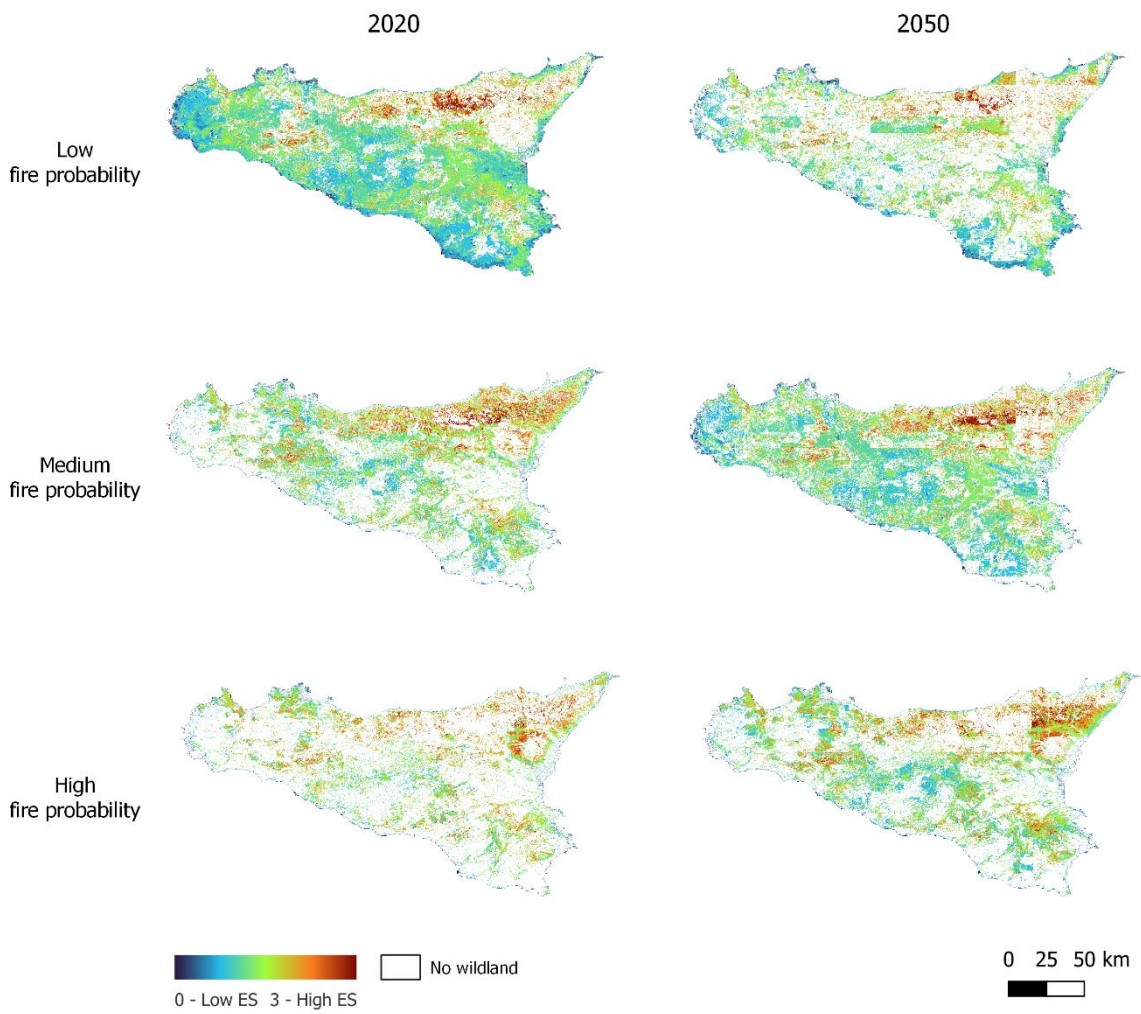


**Figure 10: Exposure map of ecological values and ES (Ecosystem Services) that may interact with levels of forest fire probability. (low, medium, and high), in 2020 and 2050.**

Linked to Figure 10, Figure 11 shows the changes (in km²) broken-down by ES (Ecosystem Services). As we observed in the exposure maps (Fig. 10), the fire hazard increases in all ES. For example, the exposure to the Carbon Mass ecosystem service and Biodiversity will increase by more than 150% in the exposed areas with high fire probability (S5, Table S3). Outdoor recreation, Soil retention, and Pollination ecosystem services will increase by 117%, 100%, and 56%, respectively. In contrast, the exposure with low fire probability will decrease between 50% and 65% each.

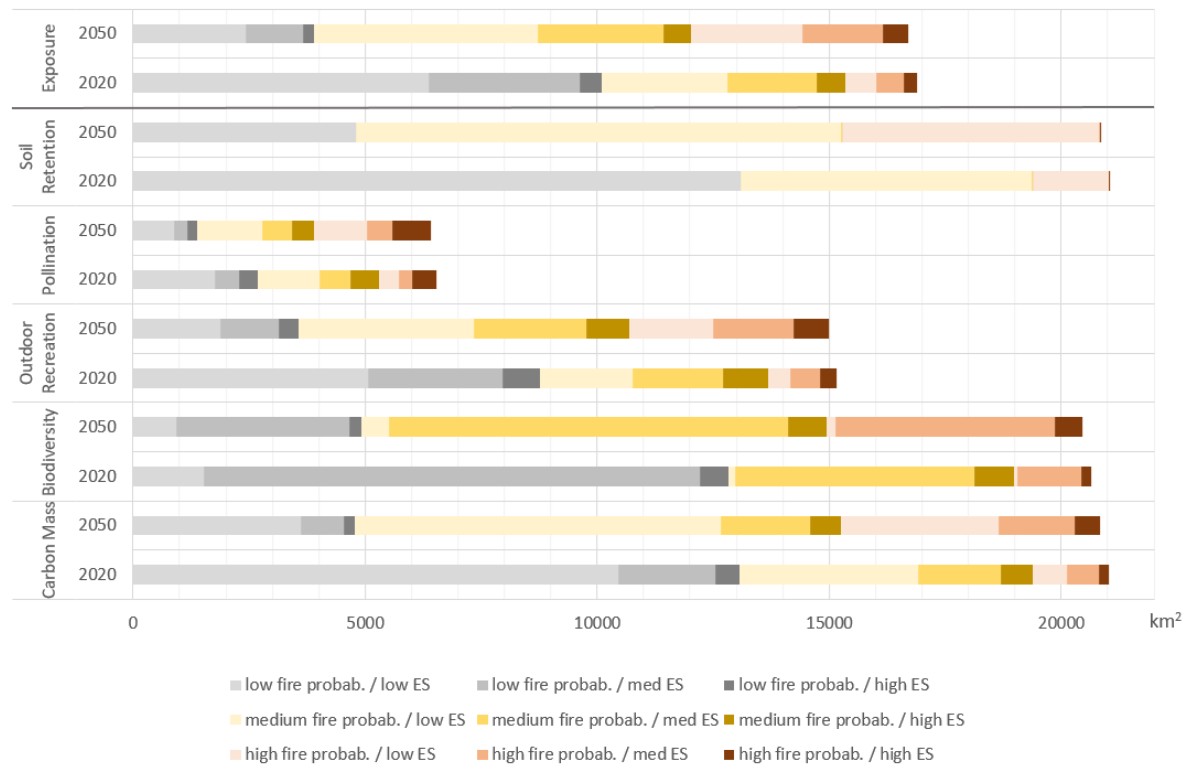

452

**Figure 11: Comparison of the fire hazard level -low (grey), medium (yellow), high (red)- by the importance of the socio-ecological elements exposed in different color tones (low, medium or high). Values show the surface average (km²) in August 2020 and 2050.**

Figure 12 shows how the percentage of vulnerable areas is distributed in each of the variables analyzed as a function of the fire probability. Therefore, following the same trend as exposed areas, ecosystem services and ecological values increase fire risk with the influence of climate change. The WUI (Wildland-Urban Interface) case, increases by 19% for high fire probabilities in 2050 and almost half of the wildfires will be at medium risk. In both WAI (Wildland-Agriculture Interface) and protected areas, half of their area could face a high fire risk in the future, doubling the 2020 data.

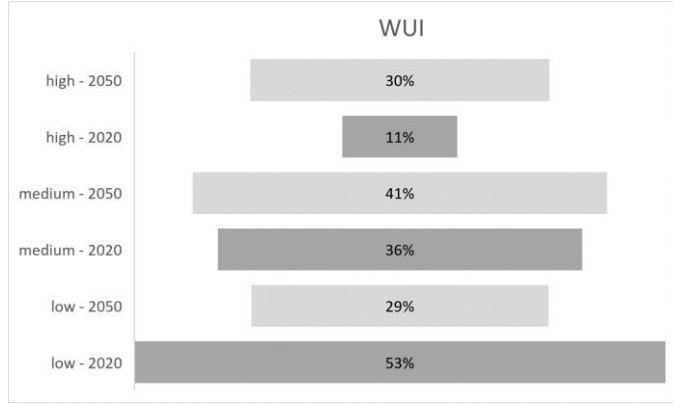


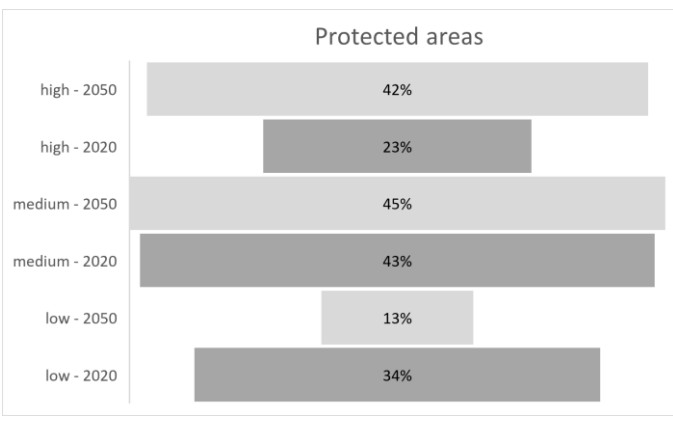

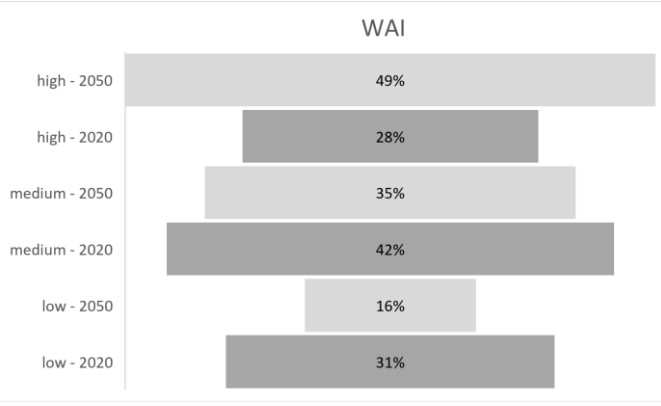

**Figure 12: Percentage of the vulnerable areas distributed in each of the variables analyzed (WUI, WAI, Protected Areas) as a**
**function of the fire probability.**
Most of the vulnerable locations close to agricultural areas have a high probability of fire. However, one of the areas with high
vulnerability in the protected area overlaps with sites that are difficult to access for the population, such as the Nebrodi Regional
Park or the Madonie Regional Natural Park (Fig. 13).
Overall, the area with the highest socio-ecological value is in the northeastern quadrant of the island, coinciding with the areas
of highest fire risk. In contrast, low-protected regions are primarily agricultural areas, urban surroundings, or areas that have
been affected by fire in the recent past. These non-vulnerability areas dominate most of the Sicilian territory.

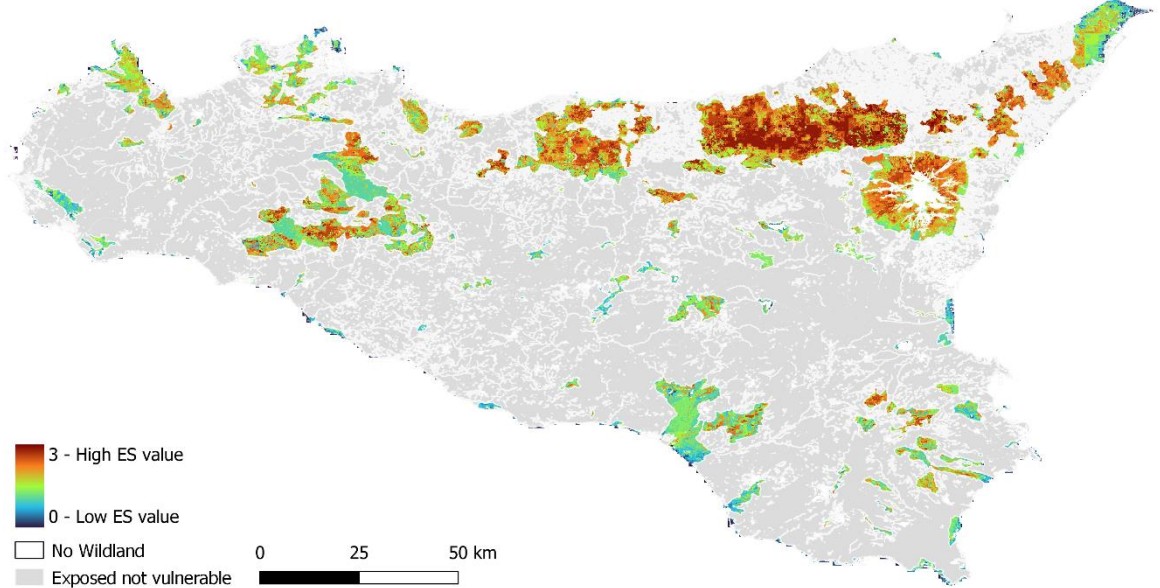


**Figure 13: Risk map of hot spots of biodiversity and ecosystem services exposed in protected areas, Wildland-Urban Interface and Wildland-Agricultural Interface in August 2020. Colored from blue with a value of 0 (low socio-environmental value) to red with a value of 3 (high socio-environmental value). Exposed but not vulnerable areas are shaded in grey. No wildland areas and no exposed are in white.**

## 4. Discussion and Summary

Although historical fire data are becoming more accessible and findable, there is still much to be done for enhancing their full use (e.g. their interoperability and reusability). The most reliable data are those collected in the field by authorized public or private institution, but in many cases, it is extremely difficult to access and download field data for the general public. In contrast, satellite data are becoming increasingly accessible. However, fires can not always be properly detected by satellites due to the following reasons: (i) they need a minimum fire size or intensity (linked to the resolution), (ii) there can be false alarms (commission errors), (iii) the information can be obscured by clouds or overstory vegetation, or the time of satellite overpass may not coincide with the fire (Hantson et al., 2013; Schroeder et al., 2008).

In this study, we use both satellite data and field data to verify and complement the fire-related information. Overall, satellite and field common problems are the scarce harmonization among data formats and the lack or bad quality of metadata. In this study, the main difficulties were the differences in parameters such as coordinate reference system, lack of metadata information and fire attributes between the yearly perimeters of fire. By integrating the data in k.LAB, all the data resources were harmonized, properly classified, and made available online with complete metadata.

Concerning the model quality, model errors are related to data location, spatio-temporal resolution or logical consistency (Guptill and Morrison, 2013; Kraak and Ormeling, 2020). Utilizing multiple data sources adds strength to the model and has been especially useful for detecting small wildfires related to land management: the vast majority of wildfires in Sicily. These kinds of wildfires may be too short-lived for the administration technicians or not intense enough to be captured by satellites. Moreover, we consider that this strategy avoided a bias in the estimation of predictors' probabilities (Roy et al., 2005).

The historical fire set was analyzed, filtered, cleaned and processed prior to fire hazard modeling. The frequency of wildfires from 2007 to 2020 was analyzed; some areas have burned more than once in the same year or more than 5 years during the 13-year period. We suggest that future studies would have to study why this phenomenon can happen and how it could be avoided, as such a high frequency of wildfires disrupts the cycle of natural processes of plants and animals, the loss of vegetation structure and composition and the associated ecosystem services.

Once the perimeters of each of the wildfires were identified, the associated information from the administration's wildfires was combined with the active fire points from the satellites to find the fire ignition area. Some differences were observed in the satellite and the government data. This may be due to reasons mentioned above: wildfires not detectable by satellites, or agricultural burnings detected as wildfires when the administration does not consider them as such. A great deal of effort was spent on data collection, cleaning, validation, pre-processing, and storage that complies with FAIR principles obtaining a reliable and open dataset: the basis of the occurrence of the fire model.

The model strength has been improved by extracting information from the predictors' data with dynamic and static variables such as meteorological or topographic data, respectively. Thus, the predictors have informed the model with values specific to each fire event. In addition, the predictors come from reliable and tested sources such as Copernicus or the Italian government as well as expert researchers and technicians. Some of the resources already existed within k.LAB such as protected areas or human settlement distribution and others were added, as fuel types or high resolution digital elevation model. The new information has been annotated in the semantic language k.IM and, like the historical fire data, now is open to any user and can interact with other k.LAB models in line with the FAIR principles.

It should be noted that this model has taken into account some of the explanatory variables at the time of ignition, but also some variables describing the ex-ante situation. Variables such as the average maximum temperature of the previous week, the accumulated precipitation or the number of days without rain were prior to the fire. The influence of climatic factors can help to predict the occurrence of wildfires related to climate change and the stress to which the forest was exposed (Halofsky et al., 2020; Trumbore et al., 2015).

The machine learning algorithm used, BN (Bayesian Network), provides a flexible and adaptable approach to structure the peculiarities of fire hazard modeling: different data sources, changes in spatio-temporal resolution and dynamic versus static

input data. BNs are useful in conducting probabilistic risk assessments since they are capable of directly modeling the whole probability distributions for stable conditions and trade-offs which is crucial for risk assessments such as fire risk of complex ecological systems. They also provide insights or quantification of the influence of a particular node on others (Kumar and Banerji, 2022). In addition, the evaluation of BNs presents much lower costs and efforts than other options, even when the dataset is partly incomplete, which is quite common for environment-related data (Bielza and Larrañaga, 2014). Most of the remaining issues are related to meteorological conditions and environmental data, either due to the punctual failure of nearby stations or problems in post-processing. However, these problems can be solved by integrating data with higher spatial resolution, which, once semantically annotated, will automatically substitute lower-quality resources.

Another advantage of BNs is that they are not a black box models: the direct interpretation of the results, based on the probabilities of the predicting variables, is given in each node probability distribution. Traditional modeling it is often difficult to access the details of the model accuracy for the end user, leading to a lack of reliability. Thanks to k.LAB and its web browser k.Explorer, the accuracy of the model is accessible and interpretable for non-expert end-users as stakeholders or land managers as we showed in the results. In line with FAIR principles, the final output and all the variables needed to compute the fire occurrence are supported by a narrative report produced at runtime to facilitate its interpretation. All these outputs are open and downloadable.

The algorithm used has provided significant values to detect areas with a high probability of fire occurrence. Thus, BNs provide a fast, reliable and accessible tool for land managers through k.LAB and semantics. The metrics related to type I and II errors can have great implications in practice, their acceptable values give credibility to the application and use of the model in real situations.

The integrated model has been able to simplify a problem as complex as the occurrence of wildfires by combining very disparate datasets. Given the results, we successfully identified the different degrees of fire hazard. The model results change according to the most influential variables that can change over time and space, such as meteorological, biophysical data and human pressure on the landscape.

By using k.LAB, a modeler can reutilize the model at any point in time, including calculating the fire hazard in real-time or in future scenarios. For example, we have run the model with future data for 2050 assuming forest management does not change. It has been analyzed how, due to extreme temperatures and the stress that they will place on vegetation, the probability of wildfires will be higher in a large part of Sicily and, therefore, new areas will be affected. The easy adaptation of the BN models together with k.Explorer visualization facilities by the stakeholders simplifies the incorporation of new data in the future to test different land management alternatives.

As the fire hazard model was incorporated into the k.LAB modeling environment,, this new model was able to interact and
connect with existing models (Villa et al., 2017). Thus, we overlapped the future fire hazard with ecosystem services that were
already developed and published by scientific researchers. We choose the ecosystems that are directly affected by fire such as
pollination, soil retention, outdoor recreation, biodiversity and carbon mass.

## 5. Conclusions

Models informing environmental decisions are usually developed in isolation, self-contained and with results mostly accessible
to code owners and their collaborators. However, in a globalized world with increasingly complex and intertwined problems,
it is key to connect knowledge and develop methods that can identify integrated solutions (Balbi et al., 2022). The application
of appropriate and reliable risk assessment techniques is key to understanding and potentially preventing future damage, but
so is making this knowledge accessible to stakeholders. This study combines the power of Artificial Intelligence and, in
particular, machine learning, knowledge representation and machine reasoning to model the risk of fire to ecosystem services
in Sicily, the largest island in the Mediterranean Sea. We used the k.LAB technology, which provides a common platform to
make data and models interoperable and accessible to non-technical users (Balbi et al., 2022).
In this study, we integrated historical fire data from 2007 to 2020 and other explanatory variables to identify the areas at the
highest risk in present and future scenarios. We developed a data-driven model using a Bayesian Network (BN) classifier.
Model analysis demonstrates that the BN algorithm applied to the historical wildfires data and their real-time variables achieves
a high range of predictive accuracy. Despite the identified limitations as the resolution of meteorological data or detect small
wildfires, the findings reveal the usefulness of the method, including the possibility to rerun the model at different time steps,
and spatial scales statically or dynamically.
The fire risk spatial results are easily accessible through a web browser that can be used freely by land managers and
stakeholders. This can help to create new prevention guidelines or focus on the risky areas. Moreover, the model gives scientists
and land managers indications about the variables that mostly affect fire probability and how they can mitigate this
environmental risk.
*Code availability.* Code used in this research is open and available at (Marquez Torres, 2023)
*Data availability.* Data used in this research is open and available at (Marquez Torres, 2023)
*Supplement.* The supplement related to this article is available on- line at: https://doi.org/10.5281/zenodo.7618466
*Author contributions*. AMT, FV, SB conceptualized the project. AMT, GS provided the data. AMT developed and calibrated
the model and ran the simulations. AMT analyzed the data and carried out the investigation. AMT visualized the data. AMT
drafted the paper. GS, SB, SK, GA reviewed and edited the paper.
*Competing interests*. The contact author has declared that none of the authors has any competing interests.
*Disclaimer*. Publisher's note: Copernicus Publications remains neutral with regard to jurisdictional claims in published maps
and institutional affiliations.
*Special issue statement*. This article is part of the special issue "The role of fire in the Earth system: understanding interactions
with the land, atmosphere, and society (ESD/ACP/BG/GMD/NHESS inter-journal SI)". It is not associated with a conference.
*Acknowledgements*: We had like to thank the University of Catania for data access, expertise and support. We acknowledge
the use of E-OBS dataset from the EU-FP6 project UERRA (https://www.uerra.eu) and the Copernicus Climate Change
Service, and the data providers in the ECA&D project (https://www.ecad.eu). We acknowledge the use of data and/or imagery
from NASA's Fire Information for Resource Management System (FIRMS) (https://earthdata.nasa.gov/firms), part of NASA's
Earth Observing System Data and Information System (EOSDIS). Map data copyrighted OpenStreetMap contributors and
available from https://www.openstreetmap.org.
Financial support: This research is part of the FPI MDM-2017-0714-18-2 funded by
MCIN/AEI/10.13039/501100011033 and partially supported by University of Catania; and by the Basque
Government through the BERC 2022-2025 program

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
