# Peer review of "Fire risk modeling: an integrated and data-driven approach applied to Sicily."

_EGUsphere, 2023_

## Referee Comment (RC1)

In this paper, authors introduce the implementation an innovative software tool to ass fire risk using both the well-known static drivers and dynamic drivers, including weather conditions before the fire event. In addition, fire hazard is evaluated considering the vulnerability of exposed elements under present and future conditions. The model has ben developed under k.LAN platform using FAIR data and resources, which makes it open and freely accessible for other researchers and stakeholders. Authors use Sicily as case study to illustrate this software implementation. Results allows to: (i) assess the relative importance of the driving variables to fire hazard; and (ii) elaborate risk maps and exposure maps for two periods, 2020 and 2050, under climate change scenarios. Different indicators for model evaluation, based on confusion matrix, are provide in the paper.

This manuscript addresses relevant scientific and technical questions within the scope of NHESS and up to international standard. The accurate description of the data, methods, experiments and computations, as well as the results obtained, allow the reproducibility of the study. Data and code have been made available on an open repository, Zenodo. English language is good and the number and quality of the references appropriate.

This paper provides a good contribution to the research in fire risk. In my opinion, it can be accepted after few revision and technical corrections listed below.

The TITLE is too generic. I propose to choose a title more focused on the present study, such as "Fire risk in Sicily: an integrated data-driven approach" or similar.

TERMONOLGY. In general, to indicate the phenomenon you often use "fire", and sometimes "wildfires" on "forest fires". Since this study focuses on unwonted fires affecting the WUI and the WAI, you should specify and use always "forest fires" (the most used in Europe).

Section 2.2 "Fire risk analyses": despite the accurate description of the three elements (hazard, vulnerability, and exposure) provided to define the risk, the type of risk you estimate in the present study is still not clear at this point. From what can be inferred in the following, you are estimating a probabilistic risk, express a probabilistic value (or likelihood) for an area to experience a fire event given certain conditions (that you can quantify) of hazard, vulnerability, and exposure. Please add few lines of description to clarify this point within section 2.2.

The quality of the FUGURES is generally very low and need to be improved. There are several errors in different figures as specified below.

- Figure 1 seems to be not correct: the histogram is not a cumulative frequency, but simply the total number of fires over the entire study period by region. The legend has to be translate in English and the font size increased to be legible. The same colormap used for the histogram should be applied to the map.
- Figure 3: I propose to move this as supplementary material and, instead, elaborate a new image to illustrate the global workflow of the methodology, from data acquisition to fire risk and exposure mapping, including model evaluation. This can also be used as graphical abstract.
- Figure 6: it's not clear since it's all black line. Please remove the administrative black borders of the municipalities.
- Figure 10: This graphic is useful only if you compere two or more models. In this case, you can simply indicate the AUC value within the text and remove the figure.
- Figure 11: move up, below Fig.5

- Figure 12: "Example of average fire occurrence in August 2020 (a) and 2050 (b)." → why you define it "average fire occurrence"? It's not a probability value? Please correct.
- Figure 14: "….in August 2018 and 2050" → I suppose that it's 2020, not 2018.
- Figure 16: "Colored from red with a value of 0 (low socio-environmental value) to blue with a value of 3 (high socioenvironmental)" → colors red and blue seems to be in the reverse order.

Somme error in Table 2:

- For the "Spatial resolution" of "Historical fire perimeter" please indicate the accuracy / minimum detectable area.
- "Temporal resulution": it's not resolution but "Time consistency". Which is the true temporal resolution? daily, monthly, yearly?  Please indicate both in the table (consistency and resolution/accuracy)
- "CRS": Indicate in full "Coordinate reference system"

Table 3 is not more informative than the description provided in the text. Please remove it or move and merge with  Table 4.

In Table 4:

- "Unite" for the Temperatures: please indicate "Celsius degrees"
- "Count of Day without Precipitation" I suppose in in # and not mm
- "Unite" for "Biomass of Forest during Fire" → you can indicate "see in (S1) Fig. S1"

Some punctual error to be fixed:

Line 28: 25**,**711 km²

Line 82: add reference and website for ARIES ([https://aries.integratedmodelling.org/](https://aries.integratedmodelling.org/))

Line 95 : no need to make a list/numbering, just simple text

Line 110 : a full stop is missing betwee « southwest Thus, »

Line 146 : « fire start and **end** date »

Line 163 : explain better the needs of "pseudo-absences" to avoid overfitting.

Line 194 : is the range for fuel type based on the flamability? please specify since it's important for the model implementation to know if it is a categorocal (just a label) or a true numerical variable.

Line 204 : the description of the BN model can be moved on a separate sub-section.

Line 235 : full stop is missing at the end of this sentence.

Line 241, with reference to S2 Table S1 : How can the max limit in the range be lower than the value for the highest bin? for example for "acc week prec" the range is 0.00-18.75 and B10 = 81.78 (but it's not the only case)

Line 160 : I suggest to rename the subsession « 2.2.2. Drivers of vulnerability **and exposed éléments** »

Line 269 : few lines to introduce AIRES are needed, as I suggested above.

Line 277 : full stop is missing at the end of this sentence.

Line 289 : 28,8814. ha

Line 228 : please explain how the model assess which is the most important variable

Line 379 : define ES here and in the figure 13

Line 456. « **T**raditional » (Upper case)

Line 489 : « from 2012 to 2019 » → correct with 2020

---

## Referee Comment (RC2)

**Major Comments:**

1. Clearly define and use the terms fire hazard, fire probability, fire danger, fire risk, fire exposure. Consider reducing the number of descriptive terms to "fire probability" and "fire risk" to avoid confusion. It is sometimes unclear for the reader to understand when the manuscript is talking about the output from the Bayesian Network model (fire probability) and when those results have been combined with ecosystem properties to determine potential impacts on ecosystems (fire risk).
- The authors could add definitions potentially around lines 90 to 100.
- Section 2.2: It is unclear exactly how the measures and properties in the table are used. The manuscript would benefit from a clearer definition here of how risk is defined and quantified.
- L277 to L281: The description of the combination of fire probability with ecosystem vulnerability would benefit from more information and clarification. As I understand it, the combination of these two products defines the fire risk.
- L369 to L376: This introduction section would benefit from more clarification. It is unclear whether the authors are talking about exposed areas in terms of fire probability or vulnerability to damage.
- L383: Should "low danger" be "low fire probability"?

2. What is the resolution of the output from the fire prediction model, for example in Figure 12?

3. The manuscript would benefit from a clearer discussion of the process of interpreting the Directed Acyclic Graphs. Some specific comments regarding result interpretation:
- Figure 4: Describe generally in the caption what all the B1 of X mean. Describe what the different colors mean in the bar charts. Describe what the arrows mean. Spell out DAG acronym in the caption.
- How and why is Figure 8 different to Figure 4?
- How do the values in Table S1 relate to Figure 8?
- Describe the connection between Figure 9 and Figure 8.
- L318-320: Please double check the values. If I am reading Figure 8 and Table S2 correctly, 18.75 should be 22.72, 25.55 should be 26.67 and 27.90 should be 29.80. Or, are the ranges associated with the bins in Figure 8 defined somewhere else rather than Table S2?
- L328 to L330: Clarify how the most influential variables are determined. Is it because they are the first connections to the fire occurrence variable in the network?
- LL397 to L402: Add in some more information about how the socioecological value was determined and what it means.

4. For future predictions – more information about which variables are available from CMIP5 sources. For instance are all variables noted in Table 5 and Fig 4 available from CMIP5 or are some variables assumed not to change (for example human settlements).

5. Are the spatial distributions of the type I and type II errors randomly distributed around Sicily or are the errors associated predominantly with one location in Sicily. The result can help interpret whether the model is more accurate for some parts of Sicily over others.

6. Section 3.2 requires an overall summary of the results found from the fire prediction in 2020 compared to 2050. For example a description of the total change in fires in the low, medium and high categories from 2020 to 2050.

7. Consider the use or "forest fires" throughout the manuscript and potentially change to "wildland fires" or "wildfires". Table S1 indicates that vegetation types such as grassland and shrubland are considered in this work, which are not necessarily forests.
* * *
**Minor Comments:**

L20: Seeing as this manuscript does not specifically study "preventing fires due to climate change" I suggest to remove this part of the sentence.

L44: Are the results in Figure 1 driven by one year (e.g. from Fig. 5, 2012 is very high), or are the results for Sicily consistently in the highest number of fire events for Italy every year 2009-2016? Also, if possible, I suggest to analyze the data for this figure from 2007 to 2020 to be consistent with the rest of the manuscript.

Figure 1: Increase font size within the image and ensure legend and axes are in English.

L55: The comment about "biodiversity is lost" is completely opposite to L22 where the introduction mentions "fires increase biodiversity". Please explain or correct this inconsistency.

Table 2: Define acronyms in the caption or a legend or table footnotes. "Temporal resolution" should be named "temporal coverage". The description says data in fire perimeter category only extends to 2019, but the study period says the research covers 2020 (L90) – please clarify.

Table 3: This one-row table seems unnecessary and could be mentioned as a sentence instead.

Table 4: Define acronyms in the caption or a legend or table footnotes. It is unclear why ARIES is a source of data in this table, because it seemed ARIES is the model used. A clearer explanation of what ARIES is and what it contains in the methodology section would be helpful.

L181: Define the acronym E-OBS.

L190: Add a reference for the DEM.

L222 to 227 and Figure 3: The authors could consider removing most of this from the manuscript, or moving to a supplement, as some of the information is repeated later.

Table 5: What was the motivation behind using the equal weight method for some variables and using the equal frequency method other variables when choosing bins? What threshold or property was evaluated to determine the method used?

L248 to L249: Do all 3 requirements need to be met to move onto the next node or only one?

L274 to L275: It is unclear what the scale "low, medium and high" refers to. What is the measure/unit underlying the calculations? Are they different for each ecosystem property, for example, is pollination the number of plants, the number of seeds or something else? It would be valuable to have a small description of these. Finally, do the distributions of the ecosystem properties support three equidistant intervals?

L291 to L293: It was unclear where this information about monthly distribution and August maximum came from until later in the manuscript. I suggest to move Figure 11 (average monthly distribution of ignitions and burned area) to directly after Figure 5, and reference it in these sentences.

Figure 7: Label axes

Figure 8: Reference Table S2 in the caption to define bin ranges.

Figure 9: Add axes labels. I suggest to spell out the fuel type on the x-axis instead of using numbers. Alternatively, add a legend for the numbers.

L350: Define "ROC".

Figure 10: Label axes.

L364 (and potentially elsewhere): Please be consistent in the use of either "medium" or "moderate" for the fire probability ranges. For example, line 364 mentions "medium", Figure 12 shows "moderate", Figure 13 shows "medium".

Figure 13: Define ES and what high and low means.

Figure 14: Caption says "green" but I think it should be "grey".

Figure 16: The description of colors in the caption is opposite to what the legend in the figure describes, i.e. caption: red = 0, blue = 3, legend: blue = 0, red = 3.

L408 to L419: This section of the discussion is confusing to understand and would benefit from a re-write.

Discussion: Consider renaming "Discussion and Summary" because some of the results are re-iterated here.

L435: What specifically is the "more complete data" that could be integrated?

Table S2: Describe what B1 to B10 are in the caption. Are these bins all determined with equal weight or equal frequency?

Table S3: Describe in the caption what ES means and the different categories. Here (and elsewhere) check the consistent use of of the full stop mark. In table S3 it is used like a "comma" in Table S2 it is used as a decimal point.
* * *
**Technical Corrections:**

Check consistency of "modelling" and "modeling", for example the title and line 15 use different versions of the spelling.

L14: Remove "fashion" – unclear meaning in this sentence.
L15: Remote Sensing is a data product not a modeling method. Therefore, I suggest to change "employing modeling methods" to "combining methods and data".
L26: ...area had a... → ...area has experience a...
L29: ...traditions, with...

L75: ...made, still few resources... → ...made, few resources...
L79: ...even within the decision support system
L111: Remove "nowadays"; ...southwest. Thus... (add the full stop)
L112: ...permanent crops. Roughly a third...
L113: ...protection, the most important being the Mount…
L230: ...values, so continuous data need to be discretized.
L240: lose less information → minimize information loss
Table 5: Firec → Fires; ount of Day… → Count of Day…
L246: ...through a heuristic search
L255: ...66% of chance → ...66% chance
L266: linked → connection
L276: ...was overlaid with wildland areas.
L297: ...once, 34.8% twice, 23.1% has burned three times or more, and nearly 6% has been burnt more than 5 times.
L364: ...level of fire occurrence…
Section 3.3 heading: Remove "and intermediate components" because it is unclear what these are.
L374: vertical axis → column
L375: axis → column
L408: remote "Concerning data sources" and start the sentence with "Although"
L409: Unclear what "fruition" means.
L456: Capitalize "Traditional".
Table S3 caption: Area of ecosystem services potentially exposed to fire...

---

## Author Comment (AC1)

**The TITLE is too generic. I propose to choose a title more focused on the present study, such as "Fire risk in Sicily: an integrated data-driven approach" or similar.**

Thanks for the suggestion, we certainly can accommodate a more specific title. We'd suggest "Fire risk modeling: an integrated data-driven approach applied to Sicily"

**TERMINOLOGY. In general, to indicate the phenomenon you often use "fire", and sometimes "wildfires" on "forest fires". Since this study focuses on unwanted fires affecting the WUI and the WAI, you should always specify and always use "forest fires" (the most used in Europe).**

We agree on the inconsistency of terminology; but we will use "wildfires" in the revised version as Reviewer 2 suggests, because we consider more vegetation types than forest, such as grassland or shrubland. This terminology is also aligned with a suggestion by Reviewer 2.

**Section 2.2 "Fire risk analyses": despite the accurate description of the three elements (hazard, vulnerability, and exposure) provided to define the risk, the type of risk you estimate in the present study is still not clear at this point. From what can be inferred in the following, you are estimating a probabilistic risk, expressing a probabilistic value (or likelihood) for an area to experience a fire event given certain conditions (that you can quantify) of hazard, vulnerability, and exposure. Please add a few lines of description to clarify this point within section 2.2.**

Thanks for the suggestion. In this manuscript we consider "Fire risk" as the potential likelihood for consequences for the elements of value in a context considering the probability of occurrence of fire hazards. Also, we consider Fire risk results from the interaction of vulnerability, exposure, and hazard. We follow the description from IPCC, 2012. We will add and clarify the description in section 2.2 and add a description of fire risk in Table 1 in the revised version.

**The quality of the FIGURES is generally very low and needs to be improved. There are several errors in different figures as specified below.**

- **Figure 1 seems to be not correct: the histogram is not a cumulative frequency, but simply the total number of fires over the entire study period by region. The legend has to be translated in English and the font size increased to be legible. The same color map used for the histogram should be applied to the map.**

  Thanks for the suggestion. We will change figure 1 to a new figure (below these lines) to be more understandable.

[Figure]

Figure 1: Total number of fires ignitions and percentage of area burned over Italy by region between 2009 and May 2016. Source: Statistics on firefighting activity, Servizi AntiIncendio Boschivo (Italian Forest Fire Services), Roma.

- **Figure 3: I propose to move this as supplementary material and, instead, elaborate a new image to illustrate the global workflow of the methodology, from data acquisition to fire risk and exposure mapping, including model evaluation. This can also be used as a graphical abstract.**

  Thanks for the suggestion, we will move Figure 3 to supplementary materials and include a global workflow in the revised version. We agree to use the global workflow as a graphical abstract

- **Figure 6: it's not clear since it's an all-black line. Please remove the administrative black borders of the municipalities.**

Thanks for the suggestion, we will change the revised version using the following image:

[Figure]

Figure 6: Fire frequency aggregated by year. The legend shows how many times the same area has been burnt during the period of 2007-2020.

- **Figure 10: This graphic is useful only if you compare two or more models. In this case, you can simply indicate the AUC value within the text and remove the figure.**

   Thanks for the suggestion, we will remove the ROC graphic and indicate the AUC value in the revised version.

- **Figure 11: move up, below Fig.5**

   Thanks for the suggestion, we will move Figure 11 up in the revised version.

- **Figure 12: "Example of average fire occurrence in August 2020 (a) and 2050 (b)." Why do you define it as "average fire occurrence"? It's not a probability value? Please correct.**

  Thanks for the suggestion, we will change the caption in the revised version as "Example of fire hazard in (a) August 2020 (b) and August 2050 classified by low, medium or high probability of fire occurrence.

- **Figure 14: "….in August 2018 and 2050" I suppose that it's 2020, not 2018.**

  Thanks for the suggestion, we will change the date in the caption in the revised version.

- **Figure 16: "Colored from red with a value of 0 (low socio-environmental value) to blue with a value of 3 (high socio-environmental)" colors red and blue seem to be in the reverse order.**

  Thanks for the suggestion, we will reverse the color order of the caption in the revised version.

Somme error in Table 2:

- **For the "Spatial resolution" of "Historical fire perimeter" please indicate the accuracy / minimum detectable area.**

  Thanks for the suggestion, the spatial resolution is less than 10 meters, as it was measured with a GPS instrument in the field. We will change in the revised version.

- **"Temporal resolution": it's not resolution but "Time consistency". Which is the true temporal resolution? daily, monthly, yearly? Please indicate both in the table (consistency and resolution/accuracy)**

  Thanks for the suggestion, we will change "Temporal resolution" to "Temporal coverage and time consistency" in the revised version. Temporal coverage is suggested by reviewer 2.

- **"CRS": Indicate in full "Coordinate reference system"**

  Thanks for the suggestion, we will indicate the full CRS as Coordinate Reference System in the revised version.

**Table 3 is not more informative than the description provided in the text. Please remove it or move and merge with Table 4.**

**In Table 4:**

- **"Unite" for the Temperatures: please indicate "Celsius degrees"**

- **"Count of Day without Precipitation" I suppose in in # and not mm**

- **"Unite" for "Biomass of Forest during Fire" you can indicate "see in (S1) Fig. S1" Some punctual error to be fixed:**

Thanks for the suggestion, we will merge Table 3 with Table 4 in the revised version. Also, we will make the changes that you suggest in Table 4 in the revised version (see the following table).

Table 4. Variables in the BN model

| Variable (semantic language) | Description | Type | Unit | Source |
|---|---|---|---|---|
| occurrence of Fire within Site | Present and absent | Discrete | 1 (fire) - 0 (no fire) | ARIES and SFI/FIRMS |
| Atmospheric Temperature | Mean temperature | Continuous | Celsius degrees | E-OBS |
| Weekly Maximum Atmospheric Temperature | Mean of maximum temperature in the last week | Continuous | Celsius degrees | ARIES (based on E-OBS data) |
| count of Day without Precipitation | Counting days since last precipitation | Continuous | # | ARIES (based on E-OBS data) |
| Weekly Precipitation Volume | Accumulated precipitation during a week | Continuous | mm | ARIES (based on E-OBS data) |
| Solar Radiation | Total solar radiation | Continuous | J/m^2 | E-OBS |
| value of Forest during Fire | Combustible biomass found in forests | Discrete | see in (S2) Fig. S2 | University of Catania |
| Elevation | Geographical elevation above sea level, as described by a digital elevation model | Continuous | m | Geoportale Regione Siciliana, Infrastruttura dati territoriali - S.I.T.R. |

| | | | | |
|---|---|---|---|---|
| Slope | Inclination of the above-water terrain in a geographical region | Continuous | grade | ARIES (based on elevation from Geoportale Regione Siciliana, Infrastruttura dati territoriali - S.I.T.R.) |
| distance to ProtectedArea | Distance to protected area | Continuous | m | ARIES (based on OSM) |
| distance to Road | Distance to road | Continuous | m | ARIES (based on OSM) |
| distance to Human Settlement | Distance to human settlement | Continuous | m | ARIES (based on OSM) |

**Line 28: 25,711 km²**

Thanks for the suggestion, we will add the punctuation in the revised version.

**Line 82: add reference and website for ARIES (https://aries.integratedmodelling.org/)**

Thanks for the suggestion, we will add in the revised version as follows:

To connect the scientific knowledge, we applied the Integrated Modeling approach of ARtificial Intelligence for Environment & Sustainability (ARIES, https://aries.integratedmodelling.org/), which is has operationalized a semantic web of accessible data, models, and other resources (Balbi et al. 2022), implementing the FAIR principles (Wilkinson et al., 2016) through the k.LAB software.

**Line 95: no need to make a list/numbering, just simple text**

Thanks for the suggestion, we will change to a simple text in the revised version.

**Line 110: a full stop is missing between « southwest Thus, »**

Thanks for the suggestion, we will add the punctuation in the revised version.

**Line 146: « fire start and end date »**

Thanks for the suggestion, we will add the "end" in the revised version.

**Line 163: explain better the needs of "pseudo-absences" to avoid overfitting.**

Thanks for the suggestion. In the revised manuscript we will add the following sentence and references:

The result of an imbalanced training dataset is a "skewed data bias" [Rennie et al 2003]. The disparity across classes will be roughly the same when training data is not skewed. The weights for the class with less training data, however, will be lower when the training data is skewed. As a consequence, classification will be unfairly biased in favor of one class over another. The learning algorithm becomes too specific, leading to overfitting [Z.li et al 2021]. Each of our predictions will be more accurate because we use more evenly distributed and balanced training data for each class, reducing the bias in our weight estimations. In consequence, our weight predictions are more reliable and our classification accuracy may increase.

Reference:

Rennie, Jason & Shih, Lawrence & Teevan, Jaime & Karger, David. (2003). Tackling the Poor Assumptions of Naive Bayes Text Classifiers. Proceedings of the Twentieth International Conference on Machine Learning. 41.

Z. Li, K. Kamnitsas and B. Glocker, "Analyzing Overfitting Under Class Imbalance in Neural Networks for Image Segmentation," in *IEEE Transactions on Medical Imaging*, vol. 40, no. 3, pp. 1065-1077, March 2021, doi: 10.1109/TMI.2020.3046692.

**Line 194: is the range for fuel type based on the flammability? please specify since it's important for the model implementation to know if it is a categorical (just a label) or a true numerical variable.**

Thanks for the question. The fuel type is categorical data defined as an identifiable association of fuel elements of distinctive species, form, size, arrangement, and continuity that will exhibit characteristic fire behavior under defined burning conditions. We will clarify in the revised version.

**Line 204: the description of the BN model can be moved on a separate subsection.**

Thanks for the suggestion, we will move under the "2.2.2. Bayesian Network model" sub-section in the revised version.

**Line 235: full stop is missing at the end of this sentence**.

Thanks for the suggestion, we will add the punctuation in the revised version.

**Line 241, with reference to S2 Table S1: How can the max limit in the range be lower than the value for the highest bin? for example for "acc week prec" the range is 0.00-18.75 and B10 = 81.78 (but it's not the only case)**

Thanks for the suggestion. The ranges are wrong in some variables, we will change the data in the revised supplementary data as follows:

| Variables | Range (min max) | Intervals | | | | | | | | | |
|---|---|---|---|---|---|---|---|---|---|---|---|
| | | B1 (min max) | B2 (min max) | B3 (min max) | B4 (min max) | B5 (min max) | B6 (min max) | B7 (min max) | B8 (min max) | B9 (min max) | B10 (min max) |
| slope (m) | 0.00 64.84 | 0.00 4.39 | 4.4 9.26 | 9.27 14.07 | 14.08 22.14 | 22.15 64.84 | | | | | |
| elevation (m) | 0.00 3138.00 | 0.00 202.05 | 202.06 350.50 | 350.51 510.50 | 510.51 713.51 | 713.52 3138.00 | | | | | |
| distance to road (m) | 0.00 4707.44 | 0.00 120.71 | 120.72 291.42 | 291.43 504.95 | 504.96 932.67 | 932.68 4707.44 | | | | | |
| fuel type | 0 7 | 0 | 1 | 2 | 3 | 4 | 5 | 6 | 7 | | |
| maximum weekly temperature (Celsius) | 1.51 39.23 | 1.51 5.29 | 5.30 9.06 | 9.07 12.83 | 12.84 16.60 | 16.61 20.37 | 20.38 24.15 | 24.16 27.92 | 27.93 31.69 | 31.70 35.46 | 35.47 39.23 |
| weekly precipitation (mm) | 0.00 125.10 | 0.00 0.05 | 0.06 2.45 | 2.46 4.75 | 4.76 7.75 | 7.76 10.85 | 10.86 14.95 | 14.96 18.75 | 18.76 25.55 | 25.56 38.45 | 38.46 125.10 |
| day without precipitation (#) | 0.00 114.0 | 0.00 2.5 | 2.5 8.5 | 18.5 35.5 | 35.5 114.0 | | | | | | |
| distance to protected area (m) | 0.00 16217.56 | 0.00 0.00 | 0.01 1014.50 | 1014.51 2582.48 | 2582.49 4859.73 | 4859.74 16217.56 | | | | | |
| distance to human (m) | 46.90 25052.21 | 46.90 3891.14 | 3891.15 6287.73 | 6287.74 8862.06 | 8862.07 12549.17 | 12549.18 25052.21 | | | | | |
| atmospheric temperature (Celsius) | - 2.96 36.55 | -2.96 0.99 | 1.00 4.94 | 4.95 8.89 | 8.90 12.84 | 12.85 16.79 | 16.80 20.75 | 20.76 24.70 | 24.71 28.65 | 28.66 32.60 | 32.61 36.55 |
| solar radiation (J/m²) | 12.00 381.00 | 12.00 85.80 | 85.81 159.60 | 159.61 233.40 | 233.41 307.20 | 307.21 381.00 | | | | | |

**Line 160: I suggest rename the subsession « 2.2.2. Drivers of vulnerability and exposed éléments »**

Thanks for the suggestion, we will rename the subsection as "2.2.3. Drivers of vulnerability and exposed elements" in the revised version because we will add the sub-section "2.2.2. Bayesian Network model" as you suggest.

**Line 269: few lines to introduce AIRES are needed, as I suggested above**.

Thanks for the suggestion, we will add a description in the Introduction section.

**Line 277: full stop is missing at the end of this sentence.**

Thanks for the suggestion, we will add the punctuation in the revised version

**Line 289: 28,8814.698 ha**

Thanks for the suggestion, we will change to 28,814.698 ha in the revised version

**Line 228: please explain how the model assess which is the most important variable**

Thanks for comment, we will explain in the revised manuscript and add the table and figures as below:

To answer which are the most influential variables of a Bayesian Network we can look at (1) the strength of influence of each edge connecting the nodes (Balbi et al. 2019) and (2) how "far", in terms of number of edges, is an input node from the final output (Marcot et al. 2006). The strength of influence is calculated from the conditional probability tables and expresses the difference between the probability distributions of two nodes by looking at the posterior probability distribution of a node, for each possible state of the parent or child node. To summarize this difference, we report normalized Euclidean distance, although other types of distances (e.g. Hellinger) are also used (Balbi et al. 2019). We show this in a new Figure 4 representing the strength of influence as the thickness of the edges. We also quantify it numerically in Table 5. The predictors with the highest strength of influence are 1. atmospheric temperature, 2. days without precipitation, 3. fuel type and 4. solar radiation (Table 5), all of which are directly linked to the final output (fire occurrence). While atmospheric temperature, number of days without precipitation, and solar radiation are expected to increase in variability and increase fire hazard with limited options for human mitigation, fuel type can be managed with punctual landscape interventions reducing its combustibility level where it is more necessary.

[Figure]

Figure 4. Directed Acyclic Graph (DAG) of the fire hazard Bayesian Network model where the thickness of the edges shows the strength of influence between nodes. Nodes show the relative probability of each variable state (Supplementary Materials, Table S2), as learned from the dataset, that leads to a fire hazard of 100%.

Table 5. Strength of influence between fire occurrence and its child nodes.

| Variable | Strength of influence |
|---|---|
| Atmospheric Temperature | 0.338 |
| Day without Precipitation | 0.193 |
| Fuel type | 0.192 |

| | |
|---|---|
| Solar Radiation | 0.191 |
| Elevation | 0.158 |
| Maximum Weekly Atmospheric Temperature | 0.154 |
| Distance to Protected Area | 0.145 |
| Slope | 0.138 |
| Distance to Road | 0.117 |
| Weekly Precipitation Volume | 0.113 |
| Distance to Human Settlement | 0.112 |

**Line 379: define ES here and in the figure 13**

Thanks for the suggestion, we will define ES as Ecosystem Services in the revised version.

**Line 456: « Traditional » (Upper case)**

Thanks for the suggestion, we will change the "t" to upper case in the revised version.

**Line 489: « from 2012 to 2019 » correct with 2020**

Thanks for the suggestion, we will correct the date in the revised version.

---

## Author Comment (AC2)

**Major Comments:**

1. **Clearly define and use the terms fire hazard, fire probability, fire danger, fire risk, fire exposure. Consider reducing the number of descriptive terms to "fire probability" and "fire risk" to avoid confusion. It is sometimes unclear for the reader to understand when the manuscript is talking about the output from the Bayesian Network model (fire probability) and when those results have been combined with ecosystem properties to determine potential impacts on ecosystems (fire risk).**

   a. **The authors could add definitions potentially around lines 90 to 100.**

      Thanks for the suggestion. We have mentioned the research queries from 95-100 and we will also highlight what we mean by fire risk in the study (based on AR6 report of IPCC, 2012). Moreover, we will provide detailed definitions of fire risk as suggested by both reviewers in the section 2.2 which is elaborated in the later comments.

   b. **Section 2.2: It is unclear exactly how the measures and properties in the table are used. The manuscript would benefit from a clearer definition here of how risk is defined and quantified.**

      Thanks for the suggestion. We will revise the manuscript and use the term 'fire risk' and 'fire hazard model' consistently and avoid using the term 'fire probability' to reduce confusion. We will revise section 2.2 in the revised version and include the definition of 'fire risk' and 'fire hazard' in Table 1. We will also briefly describe how risk is quantified in section 2.2.

   c. **L277 to L281: The description of the combination of fire probability with ecosystem vulnerability would benefit from more information and clarification. As I understand it, the combination of these two products defines the fire risk**.

      Thanks for the suggestion. We will clarify the definition in the revised version of the manuscript and highlight it in Table 1.

   d. **L369 to L376: This introduction section would benefit from more clarification. It is unclear whether the authors are talking about exposed areas in terms of fire probability or vulnerability to damage.**

      Thanks for the suggestion. We will provide a clarification in the introduction section. By 'exposed areas' we are implying the potential

area likely to be affected due to the probable occurrence of a wildfire event and consequently which socio-ecological values will be vulnerable in those exposed areas.

**L383: Should "low danger" be "low fire probability"?**

Yes, thanks for the suggestion, we will change in the revised version.

2. **What is the resolution of the output from the fire prediction model, for example in Figure 12?**

The spatial resolution is 50m, we will explain in the revised version.

3. **The manuscript would benefit from a clearer discussion of the process of interpreting the Directed Acyclic Graphs. Some specific comments regarding result interpretation:**
   a. **Figure 4: Describe generally in the caption what all the B1 of X mean. Describe what the different colors mean in the bar charts. Describe what the arrows mean. Spell out DAG acronym in the caption.**

   B1 means the first interval of each variable and the description are in the supplementary material (Table S2). The bar chart colors are randomly given. We will change the bar chart to monochromatic and reference Table S2 to the caption of Figure 4.

   b. **How and why is Figure 8 different from Figure 4?**

   Thank you for the question. Figure 4 is the distribution of the probabilities according to the fire and non-fire variables. In contrast, Figure 8 shows the distribution of probabilities only when there is fire occurrence. We will clarify in the revised version.

   c. **How do the values in Table S1 relate to Figure 8?**

   Table S1 describes the intervals of fuel type in Figure 8. We will clarify in the revised version.

   d. **Describe the connection between Figure 9 and Figure 8.**

   Figure 9 describes the relative distribution of ignition and non-ignition points by fuel type, in contrast, Figure 8 shows the Bayesian Network graph. In order to improve the ecosystem management against fires, it is more interesting to understand what types of fuels are more important when there is a fire occurrence, since the landscape is one of the few variables that we can change to reduce the fire hazard. To do this, we can directly observe the distribution of fuels type from the Bayesian Network graph. Therefore,

in the revised version we will remove Figure 9 and clarify the importance of fuels type in ecosystem management and fire prevention.

e. **L318-320: Please double check the values. If I am reading Figure 8 and Table S2 correctly, 18.75 should be 22.72, 25.55 should be 26.67 and 27.90 should be 29.80. Or, are the ranges associated with the bins in Figure 8 defined somewhere else rather than Table S2?**

Thanks for the suggestion. The ranges are wrong in some variables; we will change the data in the revised supplementary data as follows:

| Variables | Range (min max) | Intervals | | | | | | | | | |
|---|---|---|---|---|---|---|---|---|---|---|---|
| | | B1 (min max) | B2 (min max) | B3 (min max) | B4 (min max) | B5 (min max) | B6 (min max) | B7 (min max) | B8 (min max) | B9 (min max) | B10 (min max) |
| slope (m) | 0.00 64.84 | 0.00 4.39 | 4.4 9.26 | 9.27 14.07 | 14.08 22.14 | 22.15 64.84 | | | | | |
| elevation (m) | 0.00 3138.00 | 0.00 202.05 | 202.06 350.50 | 350.51 510.50 | 510.51 713.51 | 713.52 3138.00 | | | | | |
| distance to road (m) | 0.00 4707.44 | 0.00 120.71 | 120.72 291.42 | 291.43 504.95 | 504.96 932.67 | 932.68 4707.44 | | | | | |
| fuel type | 0 7 | 0 | 1 | 2 | 3 | 4 | 5 | 6 | 7 | | |
| maximum weekly temperature (Celsius) | 1.51 39.23 | 1.51 5.29 | 5.30 9.06 | 9.07 12.83 | 12.84 16.60 | 16.61 20.37 | 20.38 24.15 | 24.16 27.92 | 27.93 31.69 | 31.70 35.46 | 35.47 39.23 |
| weekly precipitation (mm) | 0.00 125.10 | 0.00 0.05 | 0.06 2.45 | 2.46 4.75 | 4.76 7.75 | 7.76 10.85 | 10.86 14.95 | 14.96 18.75 | 18.76 25.55 | 25.56 38.45 | 38.46 125.10 |
| day without precipitation (#) | 0.00 114.0 | 0.00 2.5 | 2.5 8.5 | 18.5 35.5 | 35.5 114.0 | | | | | | |

| | | | | | | | | | | | |
|---|---|---|---|---|---|---|---|---|---|---|---|
| **distance to protected area (m)** | 0.00 / 16217.56 | 0.00 / 0.00 | 0.01 / 1014.50 | 1014.51 / 2582.48 | 2582.49 / 4859.73 | 4859.74 / 16217.56 | | | | | |
| **distance to human (m)** | 46.90 / 25052.21 | 46.90 / 3891.14 | 3891.15 / 6287.73 | 6287.74 / 8862.06 | 8862.07 / 12549.17 | 12549.18 / 25052.21 | | | | | |
| **atmospheric temperature (Celsius)** | - 2.96 / 36.55 | -2.96 0.99 | 1.00 / 4.94 | 4.95 / 8.89 | 8.90 / 12.84 | 12.85 / 16.79 | 16.80 / 20.75 | 20.76 / 24.70 | 24.71 / 28.65 | 28.66 / 32.60 | 32.61 / 36.55 |
| **solar radiation (J/m²)** | 12.00 / 381.00 | 12.00 / 85.80 | 85.81 / 159.60 | 159.61 / 233.40 | 233.41 / 307.20 | 307.21 / 381.00 | | | | | |

f. **L328 to L330: Clarify how the most influential variables are determined. Is it because they are the first connections to the fire occurrence variable in the network?**

Thanks for your question. We will add the text below in the revised manuscript:

"To answer which are the most influential variables of a Bayesian Network we can look at (1) the strength of influence of each edge connecting the nodes (Balbi et al. 2019) and (2) how "far", in terms of number of edges, is an input node from the final output (Marcot et al. 2006). The strength of influence is calculated from the conditional probability tables and expresses the difference between the probability distributions of two nodes by looking at the posterior probability distribution of a node, for each possible state of the parent or child node. To summarize this difference, we report normalized Euclidean distance, although other types of distances (e.g. Hellinger) are also used (Balbi et al. 2019). We show this in a new Figure 4 representing the strength of influence as the thickness of the edges. We also quantify it numerically in Table 5. The predictors with the highest strength of influence are 1. atmospheric temperature, 2. days without precipitation, 3. fuel type and 4. solar radiation (Table 5), all of which are directly linked to the final output (fire occurrence). While atmospheric temperature, number of days without precipitation, and solar radiation are expected to increase in variability and increase fire hazard with limited options for human mitigation, fuel type can be managed with punctual landscape interventions reducing its combustibility level where it is more necessary."

[Figure]

Figure 4. Directed Acyclic Graph (DAG) of the fire hazard Bayesian network model where the thickness of the edges shows the strength of influence between nodes. Nodes show the relative probability of each variable state (Supplementary Materials, Table S2), as learned from the dataset, that leads to a fire hazard of 100%.

Table 5. Strength of influence between fire occurrence and its child nodes.

| Variable | Strength of influence |
|---|---|
| Atmospheric Temperature | 0.338 |
| Day without Precipitation | 0.193 |
| Fuel type | 0.192 |

| | |
|---|---|
| Solar Radiation | 0.191 |
| Elevation | 0.158 |
| Maximum Weekly Atmospheric Temperature | 0.154 |
| Distance to Protected Area | 0.145 |
| Slope | 0.138 |
| Distance to Road | 0.117 |
| Weekly Precipitation Volume | 0.113 |
| Distance to Human Settlement | 0.112 |

References:

Bruce G Marcot, J Douglas Steventon, Glenn D Sutherland, and Robert K McCann. 2011. Guidelines for developing and updating Bayesian belief networks applied to ecological modeling and conservation. *Canadian Journal of Forest Research*. **36**(12): 3063-3074. https://doi.org/10.1139/x06-135

Balbi, S., Selomane, O., Sitas, N., Blanchard, R., Kotzee, I., O'Farrell, P., and Villa, F.: Human dependence on natural resources in rapidly urbanising South African regions, Environ. Res. Lett., 14, 044008, https://doi.org/10.1088/1748-9326/aafe43, 2019.

g. **LL397 to L402: Add in some more information about how the socioecological value was determined and what it means.**

The socio-ecological value is based on the ecosystem services models considered (vegetation carbon mass, pollination, outdoor recreation, biodiversity and soil retention) normalized from 0 to 1, instead of others such as qualitative categorization and probabilistic approaches (normal, Poisson, binary) (Chuvieco et al., 2003). We transformed each modeling output rescaling them from 0 to 1, using the minimum and maximum value within the Sicily context. The quantitative scale was classified into 3 categories (1-low, 2-medium, 3-high) using equidistant intervals; thus integrating all

ecosystem services into a single value. We will complete the information in the revised version.

4. **For future predictions – more information about which variables are available from CMIP5 sources. For instance, are all variables noted in Table 5 and Fig 4 available from CMIP5 or are some variables assumed not to change (for example human settlements).**

Thanks for the question, as it is not clearly described in the manuscript. We forecast weather resources based on the Coupled Model Intercomparison Project 5 (CMIP5) data for RCP 8.5. We keep the other variables (solar radiation, fuel, slope, elevation, distance to road, protected area and human settlement) with the current conditions. We will clarify in the revised manuscript.

5. **Are the spatial distributions of the type I and type II errors randomly distributed around Sicily or are the errors associated predominantly with one location in Sicily. The result can help interpret whether the model is more accurate for some parts of Sicily over others.**

We have produced a map of the standard deviation of the estimated output using the learned BN model: this is because the BN model produces a probability distribution with a mean value and a standard deviation, we just didn't expose this second result in our submission. We agree it's an equally important output to show.

6. **Section 3.2 requires an overall summary of the results found from the fire prediction in 2020 compared to 2050. For example, a description of the total change in fires in the low, medium and high categories from 2020 to 2050.**

Thanks for the suggestion, we will describe the results between the two fire hazard maps in the revised version.

7. **Consider the use of "forest fires" throughout the manuscript and potentially change to "wildland fires" or "wildfires". Table S1 indicates that vegetation types such as grassland and shrubland are considered in this work, which are not necessarily forests.**

Thanks for the suggestion, we will change to "wildfire" in the revised version.

**Minor Comments:**

**L20: Seeing as this manuscript does not specifically study "preventing fires due to climate change" I suggest removing this part of the sentence.**

Thanks for the suggestion, we will change "and preventing fires due to climate change" to "and fire risk management, both under current and climate change conditions." in the revised version.

**L44: Are the results in Figure 1 driven by one year (e.g. from Fig. 5, 2012 is very high), or are the results for Sicily consistently in the highest number of fire events for Italy every year 2009-2016? Also, if possible, I suggest analyzing the data for this figure from 2007 to 2020 to be consistent with the rest of the manuscript.**

Thank you for your suggestion. Figure 1 shows the official Italian government data, which does not take into account the small fires (less than 30 ha) and only provides data for all regions from 2009 to May 2016. Figure 5 shows the analysis of the data from the government of the Sicily region checked with the data from FIRMS. After a data cleaning and analysis of both, data from the Sicilian government and FIRMS, we obtained the distribution of ignitions and burned surfaces shown in Figure 5.

**Figure 1: Increase font size within the image and ensure legend and axes are in English.**

Thanks for the suggestion, we will change the old figure to the figure below this sentence in the revised version.

[Figure]

Figure 1: Total number of fire ignitions and percentage of area burned over Italy by region between 2009 and May 2016. Source: *Statistics on firefighting activity, Servizi AntiIncendio Boschivo (Italian Forest Fire Services), Roma*.

**L55: The comment about "biodiversity is lost" is completely opposite to L22 where the introduction mentions "fires increase biodiversity". Please explain or correct this inconsistency.**

The positive or negative consequences of wildfires mainly depend on their size, intensity, and frequency. We will clarify the revised version as follows:

**Table 2: Define acronyms in the caption or a legend or table footnotes. "Temporal resolution" should be named "temporal coverage". The description says data in the fire perimeter category only extends to 2019, but the study period says the research covers 2020 (L90) – please clarify.**

Thanks for the suggestion, we will change "Temporal resolution" to "Temporal coverage and time consistency" in the revised version. Time consistency is suggested by reviewer 1 referring to the data time step. We will clarify the study period in the revised version, which covers till 2020.

**Table 3: This one-row table seems unnecessary and could be mentioned as a sentence instead.**

Thanks for the suggestion, we will remove Table 3 in the revised version.

**Table 4: Define acronyms in the caption or a legend or table footnotes. It is unclear why ARIES is a source of data in this table, because it seems that ARIES is the model used. A clearer explanation of what ARIES is and what it contains in the methodology section would be helpful.**

Thanks for the suggestion. ARtificial Intelligence for Environment & Sustainability (ARIES), is an integrated network of web accessible data, models, and other resources, implementing the FAIR principles (Wilkinson et al., 2016) through the k.LAB software, a semantic web-based integrated knowledge system. Some of the variables were integrated in the ARIES network. We will clarify what ARIES is in the introduction section. Regarding Table 4, we will define the acronyms and explain in the text the methodology followed to obtain the variables.

**L181: Define the acronym E-OBS**.

Thanks for the suggestion, we will add the acronym definition as Ensembled OBservation in the revised version.

**L190: Add a reference for the DEM**.

Thanks for the suggestion. DEM data source is Geoportale Regione Siciliana, Infrastruttura dati territoriali - S.I.T.R., we will add in the revised version.

**L222 to 227 and Figure 3: The authors could consider removing most of this from the manuscript, or moving to a supplement, as some of the information is repeated later.**

Thanks for the suggestion. We will move to supplement material and describe ARIES and k.LAB in the introduction section and k.IM language at the beginning of 2.2.1 "Fire hazard model" as below:

The model for this case study is developed using k.LAB software, a collaborative solution to achieve interoperability from the data sources to the generated modeling results. Within k.LAB an ontology-driven language called *knowledge-Integrated Modeling* (k.IM), which is grounded on an internal knowledge-based, provides the basis for the semantic annotations of resources (i.e., explicit definitions), such as external datasets, and the modeling tasks engendering traceability and knowledge integration through the lifecycle of scientific modeling (Figure S1).

Once the models are resolved, k.LAB returns spatially explicit contextualized models' results. To ensure transparency, a textual documentation of the process followed to achieve the results with annexed references and a computation workflow are also provided to the users.

**Table 5: What was the motivation behind using the equal weight method for some variables and using the equal frequency method for other variables when choosing bins? What threshold or property was evaluated to determine the method used?**

We used the most appropriate method mostly according to the data distribution of each variable and by trial and error. However, Factors to consider include the shape and spread of the data, the purpose, and level of detail of the analysis, as well as the number and size of bins. The optimal number and size of bins depend on a trade-off between information loss and information gain.

In general, Equal-width binning applied to more uniformly distributed input data as for atmospheric temperature, maximum weekly atmospheric temperature, and solar radiation.

[Figure]

For skewed distributions as for elevation, number of days without precipitation, slope, distance to protected area, distance to road, and distance to human settlement, we used Equal-frequency binning.

[Figure]

The disadvantage of Equal-frequency is that it can distort the distribution of the data and create irregular bin widths. That was the case with the "weekly precipitation" variable. After several tests, we realized that the equal-frequency produced a wrong data binning, this is the reason why we apply equal frequency in spite of its skewed distribution.

[Figure]

**L248 to L249: Do all 3 requirements need to be met to move onto the next node or only one?**

The algorithm advances when any of the conditions are met (Chen et al 2008). We will clarify this in the revised version.

**L274 to L275: It is unclear what the scale "low, medium and high" refers to. What is the measure/unit underlying the calculations? Are they different for each ecosystem property, for example, is pollination the number of plants, the number of seeds or something else? It would be valuable to have a small description of these. Finally, do the distributions of the ecosystem properties support three equidistant intervals?**

Thanks for the suggestion.

- All the models are normalized from 0 to 1 in order to compute, using the minimum and maximum value within the Sicily context.

- The ecosystem services models used are published (Martínez-López et al., 2019; Willcock et al., 2018). We briefly describe below:

    - Vegetation carbon mass: Martinez et al. 2016 calculates the above- and below-ground carbon storage in vegetation in physical units (T/ha), in accordance with Tier 1 Intergovernmental Panel on Climate Change (IPCC) methodology (IPCC, 2006; Ruesch and Gibbs, 2008).

    - Pollination: Based on land use, cropland, and weather patterns, the pollination model generates spatially explicit data of the supply and demand for insect pollination services (Martinez et al 2016).

    - Outdoor recreation: The recreation model uses ranked values to determine supply and demand and draws inspiration from the model created by Paracchini et al. (2014) for Europe. The model calculates the availability of recreational activities as a multiplicative function of naturalness and the accessibility-driven by distance of nature-based attraction parameters, computed as Euclidean distance (Martinez et al 2016)

    - Biodiversity: A Bayesian Network approach is used to spread site-based expert estimations of "biodiversity value" in order to create a map of the entire Sicilian region (Willcock, 2018).

    - Soil retention: Based on slope and contributing area, the model (Martinez et al 2016) provides biophysical estimates of soil loss and retention by plants (in tons of sediment per hectare per year) using the widely used Revised Universal Soil Loss Equation (RUSLE; Renard, 1997).

- The quantitative scale was classified into 3 categories (1-low, 2-medium, 3-high) using equidistant intervals for each socio-ecological value; thus integrating all into a single value. In this quantitative cross-assessment, the most valuable component was prioritized. The final map was overlaid with wildland areas.

We will clarify in the revised version.

**L291 to L293: It was unclear where this information about monthly distribution and August maximum came from until later in the manuscript. I suggest moving Figure 11 (average monthly distribution of ignitions and burned area) to directly after Figure 5, and reference it in these sentences.**

Thanks for the suggestion, we will move up in the revised version.

**Figure 7: Label axes**

Thanks for the suggestion, we will add the label axes in Figure 7 in the revised version.

**Figure 8: Reference Table S2 in the caption to define bin ranges.**

Thanks for the suggestion, we will reference Table S2 in Figure 8.

**Figure 9: Add axes labels. I suggest spelling out the fuel type on the x-axis instead of using numbers. Alternatively, add a legend for the numbers.**

Thanks for the suggestion, we will change the graph in order to be more understandable in the revised version as follows:

[Figure]

**L350: Define "ROC".**

Thanks for the suggestion, we will add the ROC definition as Receiver operating characteristic in the revised version.

**Figure 10: Label axes.**

Thanks for the suggestion. Reviewer 1 suggests removing Figure 10 and indicating the AUC value within the manuscript. We will follow the Reviewer 1 suggestion in the revised version.

**L364 (and potentially elsewhere): Please be consistent in the use of either "medium" or "moderate" for the fire probability ranges. For example, line 364 mentions "medium", Figure 12 shows "moderate", Figure 13 shows "medium".**

Thanks for the suggestion, we will keep "medium" both the text and figures in the revised version.

**Figure 13: Define ES and what high and low means.**

Thanks for the suggestion, we will define it in the revised version as follows:

Figure 13: Exposure map of ecological values and ES (Ecosystem Services) that may interact with levels of forest fire probability (low, medium, and high), in 2020 and 2050.

**Figure 14: Caption says "green" but I think it should be "grey".**

Thanks for the suggestion, we will change the color definition in Figure 14 in the revised version.

**Figure 16: The description of colors in the caption is opposite to what the legend in the figure describes, i.e. caption: red = 0, blue = 3, legend: blue = 0, red = 3.**

Thanks for the suggestion, we will change the color definition in the caption of the Figure 16 in the revised version.

**L408 to L419: This section of the discussion is confusing to understand and would benefit from a re- write.**

Thanks for the suggestion, we will rewrite in the revised version as follows:

Although historical fire data are becoming more accessible and findable, there is still much to be done for enhancing their full use (e.g. their interoperability and reusability). The most reliable data are those collected in the field by authorized public or private institutions, but in many cases, it is extremely difficult to access and download field data for the general public. In contrast, satellite data are becoming increasingly accessible. However, not always fire can be properly detected by satellites:

1. they need a minimum fire size or intensity (linked to the resolution),
2. there can be false alarms (commission errors),
3. they can be obscured by clouds or overstory vegetation, or

4. the time of satellite overpass may not coincide with the fire (Hantson et al., 2013, p.201; Schroeder et al., 2008).

In this study, we use both the satellite data and field data in order to verify and complement the fire-related information. Overall, satellite and field data common problems are the scarce harmonization among data formats and the lack or bad quality of metadata. In this study, the main difficulties were the differences in parameters such as coordinate reference system, lack of metadata information and fire attributes between the yearly perimeters of fire. By integrating the data in k.LAB, all the data resources were harmonized, properly classified, and made available online with complete metadata.

**Discussion: Consider renaming "Discussion and Summary" because some of the results are reiterated here.**

Thanks for the suggestion, we will rename the section name in the revised version.

**L435: What specifically is the "more complete data" that could be integrated?**

Thanks for the suggestion. "More complete data" refers to the spatial resolution of the data. We used low spatial resolution data (100m) for meteorological variables; if we compare it with other variables for which the resolution was under 50 meters. The reason was that there was no access to more detailed meteorological data at the daily time step for Sicily. We will change the sentence "more complete data" to "data with higher spatial resolution."

**Table S2: Describe what B1 to B10 are in the caption. Are these bins all determined with equal weight or equal frequency?**

We will change the caption and add more information in the Table S2 in the revised Supplementary Materials document. The type of discretization is explained in Table 5 in the manuscript.

**Table S3: Describe in the caption what ES means and the different categories. Here (and elsewhere) check the consistent use of the full stop mark. In table S3 it is used like a "comma" in Table S2 it is used as a decimal point.**

Thanks for the suggestion, we will define ES as Ecosystem Services and use the full stop mark in the revised Supplementary Material document.

**Technical Corrections:**

**Check consistency of "modelling" and "modeling", for example the title and line 15 use different versions of the spelling.**

Thanks for the suggestion, we will use "modeling" in the revised version.

**L14: Remove "fashion" – unclear meaning in this sentence.**

Thanks for the suggestion, we will remove "fashion" in the revised version.

**L15: Remote Sensing is a data product not a modeling method. Therefore, I suggest changing "employing modeling methods" to "combining methods and data".**

Thanks for the suggestion, we will use "combining methods and data" in the revised version.

**L26: ...area had a... → ...area has experience a...**

Thanks for the suggestion, we will change in the revised version.

**L29: ...traditions, with…**

Thanks for the suggestion, we will change in the revised version.

**L75: ...made, still few resources... → ...made, few resources...**

Thanks for the suggestion, we will change in the revised version.

**L79: ...even within the decision support system**

Thanks for the suggestion, we will change in the revised version.

**L111: Remove "nowadays"; ...southwest. Thus... (add the full stop)**

Thanks for the suggestion, we will change in the revised version.

**L112: ...permanent crops. Roughly a third…**

Thanks for the suggestion, we will change in the revised version.

**L113: ...protection, the most important being the Mount…**

Thanks for the suggestion, we will change in the revised version.

**L230: ...values, so continuous data need to be discretized.**

Thanks for the suggestion, we will change in the revised version.

**L240: lose less information → minimize information loss**

Thanks for the suggestion, we will change in the revised version.

**Table 5: Firec → Fires; ount of Day… → Count of Day…**

Thanks for the suggestion, we will change in the revised version.

**L246: ...through a heuristic search**

Thanks for the suggestion, we will change in the revised version.

**L255: ...66% of chance → ...66% chance**

Thanks for the suggestion, we will change in the revised version.

**L266: linked → connection**

Thanks for the suggestion, we will change in the revised version.

**L276: ...was overlaid with wildland areas.**

Thanks for the suggestion, we will change in the revised version.

**L297: ...once, 34.8% twice, 23.1% has burned three times or more, and nearly 6% has been burnt more than 5 times.**

Thanks for the suggestion, we will change in the revised version.

**L364: ...level of occurrence…**

Thanks for the suggestion, we will change in the revised version.

**Section 3.3 heading: Remove "and intermediate components" because it is unclear what these are**.

Thanks for the suggestion, we will remove it in the revised version.

**L374: vertical axis → column**

Thanks for the suggestion, we will change in the revised version.

**L375: axis → column**

Thanks for the suggestion, we will change in the revised version.

**L408: remote "Concerning data sources" and start the sentence with "Although"**

Thanks for the suggestion, we will change in the revised version.

**L409: Unclear what "fruition" means.**

Thanks for the suggestion, we will change "fruition" for "use" in the revised version.

**L456: Capitalize "Traditional".**

Thanks for the suggestion, we will change in the revised version.

**Table S3 caption: Area of ecosystem services potentially exposed to fire…**

Thanks for the suggestion, we will change in the revised version.

---

## Author Response (AR1)

**Review 1**

**The TITLE is too generic. I propose to choose a title more focused on the present study, such as "Fire risk in Sicily: an integrated data-driven approach" or similar.**

Thanks for the suggestion, we certainly can accommodate a more specific title. We have suggested "Fire risk modeling: an integrated data-driven approach applied to Sicily"

**TERMINOLOGY. In general, to indicate the phenomenon you often use "fire", and sometimes "wildfires" on "forest fires". Since this study focuses on unwanted fires affecting the WUI and the WAI, you should always specify and always use "forest fires" (the most used in Europe).**

We agree on the inconsistency of terminology; but we have used "wildfires" in the revised version as Reviewer 2 suggests, because we consider more vegetation types than forest, such as grassland or shrubland. This terminology is also aligned with a suggestion by Reviewer 2.

**Section 2.2 "Fire risk analyses": despite the accurate description of the three elements (hazard, vulnerability, and exposure) provided to define the risk, the type of risk you estimate in the present study is still not clear at this point. From what can be inferred in the following, you are estimating a probabilistic risk, expressing a probabilistic value (or likelihood) for an area to experience a fire event given certain conditions (that you can quantify) of hazard, vulnerability, and exposure. Please add a few lines of description to clarify this point within section 2.2.**

Thanks for the suggestion. In this manuscript we consider "Fire risk" as the potential likelihood for consequences for the elements of value in a context considering the probability of occurrence of fire hazards. Also, we consider Fire risk results from the interaction of vulnerability, exposure, and hazard. We follow the description from IPCC, 2012. We have added and clarify the description in section 2.2 and have added a description of fire risk in Table 1 in the revised version.

Lines: 124, 139-140, (revised version), 430, 498-499 (track changes manuscript).

**The quality of the FIGURES is generally very low and needs to be improved. There are several errors in different figures as specified below.**

- **Figure 1 seems to be not correct: the histogram is not a cumulative frequency, but simply the total number of fires over the entire study period by region. The legend has to be translated in English and the font**

**size increased to be legible. The same colormap used for the histogram should be applied to the map.**

Thanks for the suggestion. We have changed figure 1 to a new figure to be more understandable.

Lines: 52 (revised version), 141 (track changes manuscript).

- **Figure 3: I propose to move this as supplementary material and, instead, elaborate a new image to illustrate the global workflow of the methodology, from data acquisition to fire risk and exposure mapping, including model evaluation. This can also be used as a graphical abstract.**

Thanks for the suggestion, we have moved Figure 3 to supplementary materials (S1, Fig. S2) and include a global workflow (S1, Fig. S1). We agree to use the global workflow as a graphical abstract.

Lines: 142, 148, 250, 254 (revised version), 501, 507, 1017, 1021 (track changes manuscript).

- **Figure 6: it's not clear since it's an all-black line. Please remove the administrative black borders of the municipalities.**

Thanks for the suggestion, we have changed in the revised version.

Lines: 360 (revised version), 1502 (track changes manuscript).

- **Figure 10: This graphic is useful only if you compare two or more models. In this case, you can simply indicate the AUC value within the text and remove the figure.**

Thanks for the suggestion, we have removed the ROC graphic and indicate the AUC value in the revised version.

Lines: 410-412 (revised version), 1758-1760 (track changes manuscript).

- **Figure 11: move up, below Fig.5**

Thanks for the suggestion, we have moved Figure 11 up in the revised version.

Figure 11 has changed to Figure 5, lines: 352 (revised version), 1491 (track changes manuscript).

- **Figure 12: "Example of average fire occurrence in August 2020 (a) and 2050 (b)." Why do you define it as "average fire occurrence"? It's not a probability value? Please correct.**

  Thanks for the suggestion, we have changed the caption in the revised version as "Example of fire hazard in (a) August 2020 (b) and August 2050 classified by low, medium or high probability of fire occurrence and the colors to be more understandable.

  Figure 12 has changed to Figure 9, lines: 423 (revised version), 1860 (track changes manuscript).

- **Figure 14: "….in August 2018 and 2050" I suppose that it's 2020, not 2018.**

  Thanks for the suggestion, we have changed the date in the caption in the revised version.

  Figure 14 has changed to Figure 11, lines: 449 (revised version), 2023 (track changes manuscript).

- **Figure 16: "Colored from red with a value of 0 (low socio-environmental value) to blue with a value of 3 (high socio-environmental)" colors red and blue seem to be in the reverse order.**

  Thanks for the suggestion, we have reversed the color order of the caption in the revised version.

  Figure 16 has changed to Figure 13, lines: 469 (revised version), 2052 (track changes manuscript).

**Some error in Table 2:**

- **For the "Spatial resolution" of "Historical fire perimeter" please indicate the accuracy / minimum detectable area.**

- **"Temporal resolution": it's not resolution but "Time consistency". Which is the true temporal resolution? daily, monthly, yearly? Please indicate both in the table (consistency and resolution/accuracy)**

- **"CRS": Indicate in full "Coordinate reference system"**

Thanks for the suggestion, the spatial resolution is less than 10 meters, as it was measured with a GPS instrument in the field. We have changed in the revised version.

We have changed "Temporal resolution" to "Temporal coverage and time consistency" in the revised version. Temporal coverage is suggested by reviewer 2.

We have indicated the full CRS as Coordinate Reference System in the revised version.

Table 2, lines: 160 (revised version), 519 (track changes manuscript).

**Table 3 is not more informative than the description provided in the text. Please remove it or move and merge with Table 4.**

**In Table 4:**

- **"Unite" for the Temperatures: please indicate "Celsius degrees"**

- **"Count of Day without Precipitation" I suppose in in # and not mm**

- **"Unite" for "Biomass of Forest during Fire" you can indicate "see in (S1) Fig. S1" Some punctual error to be fixed:**

Thanks for the suggestion, we have merged Table 3 with Table 4 in the revised version. Also, we have made the changes that you suggest in Table 4 in the revised version.

Table 4 has changed to Table 3, lines 191 (revised version), 769 (track changes manuscript).

**Line 28: 25,711 km²**

Thanks for the suggestion, we have added the punctuation in the revised version.

Lines: 29 (revised version and track changes manuscript).

**Line 82: add reference and website for ARIES (https://aries.integratedmodelling.org/)**

Thanks for the suggestion, we have added in the revised version.

Lines: 85 (revised version), 266 (track changes manuscript).

**Line 95: no need to make a list/numbering, just simple text**

Thanks for the suggestion, we have changed to a simple text in the revised version.

Lines: 98-99 (revised version), 279-280 (track changes manuscript).

**Line 110: a full stop is missing between « southwest Thus, »**

Thanks for the suggestion, we have added the punctuation in the revised version.

Lines: 114 (revised version), 420 (track changes manuscript).

**Line 146: « fire start and end date »**

Thanks for the suggestion, we have added the "end" in the revised version.

Lines: 161 (revised version), 664 (track changes manuscript).

**Line 163: explain better the needs of "pseudo-absences" to avoid overfitting.**

Thanks for the suggestion. In the revised manuscript we have added the new sentences and references:

Lines: 177-180 (revised version), 680-683 (track changes manuscript).

**Line 194: is the range for fuel type based on the flammability? please specify since it's important for the model implementation to know if it is a categorical (just a label) or a true numerical variable.**

Thanks for the question. The fuel type is categorical data defined as an identifiable association of fuel elements of distinctive species, form, size, arrangement, and continuity that will exhibit characteristic fire behavior under defined burning conditions. We have clarified in the revised version.

Lines: 220-223 (revised version), 934-937 (track changes manuscript).

**Line 204: the description of the BN model can be moved on a separate subsection.**

Thanks for the suggestion, we have moved under the "2.2.2. Bayesian Network model" sub-section in the revised version.

Lines: 232 (revised version), 999 (track changes manuscript).

**Line 235: full stop is missing at the end of this sentence**.

Thanks for the suggestion, we have added the punctuation in the revised version.

Lines: 261 (revised version), 1028 (track changes manuscript).

**Line 241, with reference to S2 Table S1: How can the max limit in the range be lower than the value for the highest bin? for example for "acc week prec" the range is 0.00-18.75 and B10 = 81.78 (but it's not the only case)**

Thanks for the suggestion. The ranges are wrong in some variables; we have changed the data in the revised supplementary material:

S2 Table S1 changes to S4, Table S2, lines: 265-267 (revised version), 1100-1102 (track changes manuscript).

**Line 160: I suggest rename the subsession « 2.2.2. Drivers of vulnerability and exposed éléments »**

Thanks for the suggestion, we have renamed the subsection as "2.2.3. Drivers of vulnerability and exposed elements" in the revised version because we have added the sub-section "2.2.2. Bayesian Network model", as you suggest.

Lines: 300 (revised version), 1234 (track changes manuscript).

**Line 269: few lines to introduce AIRES are needed, as I suggested above**.

Thanks for the suggestion, we have added a description in the Introduction section.

Lines: 84-86 (revised version), 265-267 (track changes manuscript).

**Line 277: full stop is missing at the end of this sentence.**

Thanks for the suggestion, we have added the punctuation in the revised version.

Lines: 328 (revised version), 1344 (track changes manuscript).

**Line 289: 28,8814.698 ha**

Thanks for the suggestion, we have changed to 28,814.698 ha in the revised version.

Lines: 342 (revised version), 1358 (track changes manuscript).

**Line 228: please explain how the model assess which is the most important variable.**

Thanks for the suggestion, we have added more information in the revised version.

Lines: 280-291 (revised version), 1170-1181 (track changes manuscript).

**Line 379: define ES here and in the Figure 13**

Thanks for the suggestion, we have defined ES as Ecosystem Services in the revised version.

Figure 13 has changed to Figure 10, lines: 442 (revised version), 1990 (track changes manuscript).

**Line 456. « Traditional » (Upper case)**

Thanks for the suggestion, we have changed the "t" to upper case in the revised version.

Lines: 525 (revised version), 2263 (track changes manuscript).

**Line 489: « from 2012 to 2019 » correct with 2020**

Thanks for the suggestion, we have corrected the date in the revised version.

Lines: 558 (revised version), 2322 (track changes manuscript).

**Review 2**

**Major Comments:**

1. **Clearly define and use the terms fire hazard, fire probability, fire danger, fire risk, fire exposure. Consider reducing the number of descriptive terms to "fire probability" and "fire risk" to avoid confusion. It is sometimes unclear for the reader to understand when the manuscript is talking about the output from the Bayesian Network model (fire probability) and when those results have been combined with ecosystem properties to determine potential impacts on ecosystems (fire risk).**

   a. **The authors could add definitions potentially around lines 90 to 100.**

      Thanks for the suggestion. We have mentioned the research queries and we have also highlight what we mean by fire risk in the study (based on AR6 report of IPCC, 2012). Moreover, we have provided detailed definitions of fire risk as suggested by both reviewers in the section 2.2 which is elaborated in the later comments.

      Lines: 95-100 (revised version), 276-280 (track changes manuscript).

   b. **Section 2.2: It is unclear exactly how the measures and properties in the table are used. The manuscript would benefit from a clearer definition here of how risk is defined and quantified.**

      Thanks for the suggestion. We have revise the manuscript and use the term 'fire risk' and 'fire hazard model' consistently and avoid using the term 'fire probability' to reduce confusion. We have revise section 2.2 in the revised version and include the definition of 'fire risk' and 'fire hazard' in Table 1. We have also briefly described how risk is quantified in section 2.2.

      Lines: 122-124 and 139-143 (revised version), 427-429 and 497-501 (track changes manuscript).

   c. **L277 to L281: The description of the combination of fire probability with ecosystem vulnerability would benefit from more information and clarification. As I understand it, the combination of these two products defines the fire risk.**

      Thanks for the suggestion. We have clarified the definition in the revised version of the manuscript and highlight it in Table 1.

Lines: 122-124, 139-143, 334-336 (new manuscript), 427-429, 497-501 and 1350-1351 (track changes manuscript).

d. **L369 to L376: This introduction section would benefit from more clarification. It is unclear whether the authors are talking about exposed areas in terms of fire probability or vulnerability to damage.**

Thanks for the suggestion. We have provided a clarification in the introduction section and Table 1. By 'exposed areas' we are implying the potential area likely to be affected due to the probable occurrence of a wildfire event and consequently which ecosystem services and biodiversity will be vulnerable in those exposed areas.

Lines: 433 (new manuscript), 1870 (track changes manuscript).

**L383: Should "low danger" be "low fire probability"?**

Yes, thanks for the suggestion, we have changed in the revised version.

Lines: 448 (revised version), 1996 (track changes manuscript).

2. **What is the resolution of the output from the fire prediction model, for example in Figure 12?**

The spatial resolution is 50m, we have explained in the revised version.

Lines: 422 and 433 (revised version), 1770 and 1870 (track changes manuscript).

3. **The manuscript would benefit from a clearer discussion of the process of interpreting the Directed Acyclic Graphs. Some specific comments regarding result interpretation:**
   a. **Figure 4: Describe generally in the caption what all the B1 of X mean. Describe what the different colors mean in the bar charts. Describe what the arrows mean. Spell out DAG acronym in the caption.**

B1 means the first interval of each variable and the description are in the supplementary material (S4, Table S2).

Lines: 262-267 (revised version), 1097-1103 (track changes manuscript).

The bar chart colors are randomly given. We have changed the bar chart to monochromatic, have added DAG acronym and have references S4, Table S2 to the caption of Figure 4.

DAG acronym in the text, lines: 242, (revised version), 1009 (track changes manuscript).

Figure 4 changed to Figure 3, lines 293 (revised version), 1202 (track changes manuscript).

**b. How and why is Figure 8 different from Figure 4?**

Thank you for the question. Figure 4 is the distribution of the probabilities according to the fire and non-fire variables. In contrast, Figure 8 shows the distribution of probabilities only when there is fire occurrence. We have clarified in the revised version.

Lines 278-280, 370-373 (revised version), 1168-1170, 1559-1562 (track changes manuscript).

**c. How do the values in Table S1 relate to Figure 8?**

Table S1 describes the intervals of fuel type in Figure 8. We have clarified in the caption in the revised version.

Figure 8 has changed to Figure 3 and Table S1 to S4, table S2, lines: 293 (revised version), 1208 (track changes manuscript).

**d. Describe the connection between Figure 9 and Figure 8.**

Figure 9 describes the relative distribution of ignition and non-ignition points by fuel type, in contrast, Figure 8 shows the Bayesian Network graph. In order to improve the ecosystem management against fires, it is more interesting to understand what types of fuels are more important when there is a fire occurrence, since the landscape is one of the few variables that we can change to reduce the fire hazard. To do this, we can directly observe the distribution of fuels type from the Directed Acyclic Graph (DAG) of the Bayesian Network (Figure 3 in the revised version). Therefore, in the revised version we have removed Figure 9 and clarify the importance of fuels type in ecosystem management and fire prevention.

Lines: 221-223 and 288-291 (revised version), 935-939 and 1177-1181 (track changes manuscript).

**e. L318-320: Please double check the values. If I am reading Figure 8 and Table S2 correctly, 18.75 should be 22.72, 25.55 should be 26.67 and 27.90 should be 29.80. Or, are the ranges associated with the bins in Figure 8 defined somewhere else rather than Table S2?**

Thanks for the suggestion. The ranges are wrong in some variables; we have changed the data in the revised supplementary data (S4, Table S2).

 f. **L328 to L330: Clarify how the most influential variables are determined. Is it because they are the first connections to the fire occurrence variable in the network?**

Thanks for your question. We have explained in the revised version.

Lines: 282-291 (revised version), 1155-1166 (track changes manuscript).

 g. **LL397 to L402: Add in some more information about how the socioecological value was determined and what it means.**

Thanks for the suggestion, we have clarified in the revised version.

Lines: 301-328 (revised version), 1227-1336 (track changes manuscript).

4. **For future predictions – more information about which variables are available from CMIP5 sources. For instance, are all variables noted in Table 5 and Fig 4 available from CMIP5 or are some variables assumed not to change (for example human settlements).**

Thanks for the question, as it is not clearly described in the manuscript. We forecast weather resources based on the Coupled Model Intercomparison Project 5 (CMIP5) data for RCP 8.5. We keep the other variables (solar radiation, fuel, slope, elevation, distance to road, protected area and human settlement) with the current conditions. We have clarified in the revised manuscript.

Lines: 334-339 (revised version), 1342-1347 (track changes manuscript).

5. **Are the spatial distributions of the type I and type II errors randomly distributed around Sicily or are the errors associated predominantly with one location in Sicily. The result can help interpret whether the model is more accurate for some parts of Sicily over others.**

We have produced a map of the standard deviation of the estimated output using the learned BN model: this is because the BN model produces a probability distribution with a mean value and a standard deviation, we just didn't expose this second result in our submission. We agree it's an equally important output to show.

Lines: 413-415 (revised version), 1753-1755 (track changes manuscript).

6. **Section 3.2 requires an overall summary of the results found from the fire prediction in 2020 compared to 2050. For example, a description of the total change in fires in the low, medium and high categories from 2020 to 2050.**

Thanks for the suggestion, we have described the results between the two fire hazard maps in the revised version.

Lines: 425-431 (revised version), 1854-1860 (track changes manuscript).

7. **Consider the use of "forest fires" throughout the manuscript and potentially change to "wildland fires" or "wildfires". Table S1 indicates that vegetation types such as grassland and shrubland are considered in this work, which are not necessarily forests.**

Thanks for the suggestion, we have changed to "wildfire" in the revised version.

**Minor Comments:**

**L20: Seeing as this manuscript does not specifically study "preventing fires due to climate change" I suggest removing this part of the sentence.**

Thanks for the suggestion, we have changed "and preventing fires due to climate change" to "and fire risk management, both under current and climate change conditions." in the revised version.

Lines: 21 (revised version), 21 (track changes manuscript).

**L44: Are the results in Figure 1 driven by one year (e.g. from Fig. 5, 2012 is very high), or are the results for Sicily consistently in the highest number of fire events for Italy every year 2009-2016? Also, if possible, I suggest analyzing the data for this figure from 2007 to 2020 to be consistent with the rest of the manuscript.**

Thank you for your suggestion. Figure 1 shows the official Italian government data downloadable on their webpage, which does not take into account the small fires (less than 30 ha) and only provides data for all regions from 2009 to May 2016. This is the only data provided by the Italian government for all regions, as each region obtains the data and, depending on the region, provides it to the government. Some regions have on their government websites their data, although not all for the same time period. We thought that, although the time period is different from the one analyzed for Sicily, it was important to show that Sicily is an important region to analyze because the proportion of fires and burned areas is much higher than the other Italian regions. We have clarified it in the caption of Figure 1 (lines: 52 (revised version), 140 (track changes manuscript).

Figure 5 shows the analysis of the data from the government of the Sicily region (area burned detected less than 1ha) checked with the data from FIRMS. After a data cleaning and analysis of both, data from the Sicilian government and FIRMS, we obtained the distribution of ignitions and burned surfaces shown in Figure 5 (Figure

4 in the revised version, lines: 350 (revised version), 1487 (track changes manuscript).

**Figure 1: Increase font size within the image and ensure legend and axes are in English.**

Thanks for the suggestion, we have changed the old figure to a new figure in the revised version.

Lines: 52 (revised version), 140 (track changes manuscript).

**L55: The comment about "biodiversity is lost" is completely opposite to L22 where the introduction mentions "fires increase biodiversity". Please explain or correct this inconsistency.**

The positive or negative consequences of wildfires mainly depend on their size, intensity, and frequency. We have clarified the revised version.

Lines: 57 (revised version), 145 (track changes manuscript).

**Table 2: Define acronyms in the caption or a legend or table footnotes. "Temporal resolution" should be named "temporal coverage". The description says data in the fire perimeter category only extends to 2019, but the study period says the research covers 2020 (L90) – please clarify.**

Thanks for the suggestion, we have changed "Temporal resolution" to "Temporal coverage and time consistency" in the revised version (Table 2). Time consistency is suggested by reviewer 1 referring to the data time step. We have clarified the study period in the revised version, which covers till 2020.

Lines: 160 (revised version), 522 (track changes manuscript).

**Table 3: This one-row table seems unnecessary and could be mentioned as a sentence instead.**

Thanks for the suggestion, we have joined both tables.

Table 3 and Table 4 have changed to Table 3, lines: 191 (revised version), 772 (track changes manuscript).

**Table 4: Define acronyms in the caption or a legend or table footnotes. It is unclear why ARIES is a source of data in this table, because it seems that ARIES is the model used. A clearer explanation of what ARIES is and what it contains in the methodology section would be helpful.**

Thanks for the suggestion, we have clarified what ARIES is in the introduction section.

Lines: 84-87, 100 (revised version), 268-270, 409 (track changes manuscript).

Regarding Table 4 (Table 3 in the revised version), we have defined the acronyms as a footnotes and have explained the methodology followed.

Lines: 191-206 (revised version), 772-923 (track changes manuscript).

**L181: Define the acronym E-OBS**.

Thanks for the suggestion, we have added the acronym definition as Ensembled OBservation in the revised version.

Lines: 189-190 and 195 (revised version), 770-771 and 854 (track changes manuscript).

**L190: Add a reference for the DEM**.

Thanks for the suggestion. DEM data source is SITR: *Sistema Informativo Territoriale Regionale* (Regional Spatial Information System), we have added in the revised version.

Table 3, lines: 196 (revised version), 855 (track changes manuscript).

**L222 to 227 and Figure 3: The authors could consider removing most of this from the manuscript, or moving to a supplement, as some of the information is repeated later.**

Thanks for the suggestion. We have moved to supplement material.

S1 Fig. S2 in Supplementary material and lines: 148, 250-254 (revised version), 510,1005-1010 (track changes manuscript).

**Table 5: What was the motivation behind using the equal weight method for some variables and using the equal frequency method for other variables when choosing bins? What threshold or property was evaluated to determine the method used?**

We used the most appropriate method mostly according to the data distribution of each variable and by trial and error. However, factors to consider include the shape and spread of the data, the purpose, and level of detail of the analysis, as well as the number and size of bins. The optimal number and size of bins depend on a trade-off between information loss and information gain. We have explained in Supplementary Material S3.

**L248 to L249: Do all 3 requirements need to be met to move onto the next node or only one?**

The algorithm advances when any of the conditions are met (Chen et al 2008). We have clarified this in the revised version.

Lines: 274-277 (revised version), 1097,1153-1155 (track changes manuscript).

**L274 to L275: It is unclear what the scale "low, medium and high" refers to. What is the measure/unit underlying the calculations? Are they different for each ecosystem property, for example, is pollination the number of plants, the number of seeds or something else? It would be valuable to have a small description of these. Finally, do the distributions of the ecosystem properties support three equidistant intervals?**

Thanks for the suggestion.

- All the models are normalized from 0 to 1 in order to compute, using the minimum and maximum value within the Sicily context.
- The ecosystem services models used are published (Martínez-López et al., 2019; Willcock et al., 2018). We briefly describe in the revised version.
- The quantitative scale was classified into 3 categories (1-low, 2-medium, 3-high) using equidistant intervals for each socio-ecological value; thus integrating all into a single value. In this quantitative cross-assessment, the most valuable component was prioritized. The final map was overlaid with wildland areas.

Lines: 301-328 (revised version), 1230-1339 (track changes manuscript).

**L291 to L293: It was unclear where this information about monthly distribution and August maximum came from until later in the manuscript. I suggest moving Figure 11 (average monthly distribution of ignitions and burned area) to directly after Figure 5, and reference it in these sentences.**

Thanks for the suggestion, we have moved up in the revised version.

Figure 11 change to Figure 5, lines: 353 (revised version), 1487 (trach changes manuscript).

References, lines: 345, 417 (revised version), 1356, 1760 (track changes manuscript).

**Figure 7: Label axes**

Thanks for the suggestion, we have added the label axes in Figure 7 in the revised version. In addition, we have changed the colors to make them more readable and have added more information in the caption.

Lines: 367 (revised version), 1545 (track changes manuscript).

**Figure 8: Reference Table S2 in the caption to define bin ranges.**

Thanks for the suggestion, we have referenced in the revised version.

S2, Table S2 has changed to S4, Table S2 in the revised version of the supplementary materials.

**Figure 9: Add axes labels. I suggest spelling out the fuel type on the x-axis instead of using numbers. Alternatively, add a legend for the numbers.**

Thanks for the suggestion, we have added the axes labels at Figure 3, in the Fuel type node (line 292 in the revised version and 1196 in the manuscript with track changes). We have removed Figure 9 as the information is repeated.

**L350: Define "ROC".**

Thanks for the suggestion, we have added the ROC definition as Receiver Operating Characteristic in the revised version.

Lines: 407 (revised version), 1750 (track changes manuscript).

**Figure 10: Label axes.**

Thanks for the suggestion. Reviewer 1 suggests removing Figure 10 and indicating the AUC value within the manuscript. We have followed the Reviewer 1 suggestion in the revised version.

Lines: 411-412 (revised version), 1754-1755 (track changes manuscript).

**L364 (and potentially elsewhere): Please be consistent in the use of either "medium" or "moderate" for the fire probability ranges. For example, line 364 mentions "medium", Figure 12 shows "moderate", Figure 13 shows "medium".**

Thanks for the suggestion, we have kept "medium" in both the text and figures in the revised version.

Figure 12 has changed to Figure 9 in the revised version (line 424) and the track changes manuscript (line 1856).

**Figure 13: Define ES and what high and low means.**

Thanks for the suggestion, we have defined it in the revised version.

Figure 13 has changed to Figure 10 in the revised version, lines: 442-443 (revised version) 1985-1986 (track changes manuscript).

**Figure 14: Caption says "green" but I think it should be "grey".**

Thanks for the suggestion, we have changed the color definition in Figure 14 in the revised version.

Figure 14 has changed to Figure 11 in the revised version, lines: 450-451 (revised version), 2019-2020 (track changes manuscript).

**Figure 16: The description of colors in the caption is opposite to what the legend in the figure describes, i.e. caption: red = 0, blue = 3, legend: blue = 0, red = 3.**

Thanks for the suggestion, we have changed the color definition in the caption of the Figure 16 in the revised version.

Figure 16 has changed to Figure 13 in the revised version, lines: 469-472 (revised version), 2047-2050 (track changes manuscript).

**L408 to L419: This section of the discussion is confusing to understand and would benefit from a re- write.**

Thanks for the suggestion, we have rewritten the revised version.

Lines: 474-485 (revised version), 2052-2063 (track changes manuscript).

**Discussion: Consider renaming "Discussion and Summary" because some of the results are reiterated here.**

Thanks for the suggestion, we have renamed in the revised version.

Lines: 473 (revised version), 2051 (track changes manuscript).

**L435: What specifically is the "more complete data" that could be integrated?**

Thanks for the suggestion. "More complete data" refers to the spatial resolution of the data. We used low spatial resolution data (100m) for meteorological variables; if we compare it with other variables for which the resolution was under 50 meters. The reason was that there was no access to more detailed meteorological data at the daily time step for Sicily. We have changed the sentence "more complete data" to "data with higher spatial resolution."

Lines: 522 (revised version), 2255 (track changes manuscript).

**Table S2: Describe what B1 to B10 are in the caption. Are these bins all determined with equal weight or equal frequency?**

We have changed the caption, the type of discretization is shown in Table 4 in the revised version and the manuscript with track changes.

Table 3 has changed to Table 4 in the revised version (line 268) and manuscript with track changes (line 1091).

We have added more information regarding to bins discretization in the section S3, on the revised Supplementary Materials document.

Lines: 262-263 (revised version) and 1085-1086 (track changes manuscript)

**Table S3: Describe in the caption what ES means and the different categories. Here (and elsewhere) check the consistent use of the full stop mark. In table S3 it is used like a "comma" in Table S2 it is used as a decimal point.**

Thanks for the suggestion, we have defined ES as Ecosystem Services and use the full stop mark as a decimal point and a comma as thousand separator in the revised Supplementary Material document.

S3, Table S3 has changed to S5, Table S3.

**Technical Corrections:**

**Check consistency of "modelling" and "modeling", for example the title and line 15 use different versions of the spelling.**

Thanks for the suggestion, we have used "modeling" in the revised version.

**L14: Remove "fashion" – unclear meaning in this sentence.**

Thanks for the suggestion, we have removed "fashion" in the revised version.

Line: 15 (revised version and track changes manuscript).

**L15: Remote Sensing is a data product not a modeling method. Therefore, I suggest changing "employing modeling methods" to "combining methods and data".**

Thanks for the suggestion, we have used "combining methods and data" in the revised version.

Line: 16 (revised version and track changes manuscript).

**L26: ...area had a... → ...area has experience a...**

Thanks for the suggestion, we have changed in the revised version.

Lines: 27-28 (revised version and track changes manuscript).

**L29: ...traditions, with…**

Thanks for the suggestion, we have changed in the revised version.

Line: 30 (revised version and track changes manuscript).

**L75: ...made, still few resources... → ...made, few resources...**

Thanks for the suggestion, we have changed in the revised version.

Lines: 78 (revised version), 262 (track changes manuscript).

**L79: ...even within the decision support system**

Thanks for the suggestion, we have changed in the revised version.

Lines: 82 (revised version), 266 (track changes manuscript).

**L111: Remove "nowadays"; ...southwest. Thus... (add the full stop)**

Thanks for the suggestion, we have changed in the revised version.

Lines: 114 (revised version), 424 (track changes manuscript).

**L112: ...permanent crops. Roughly a third…**

Thanks for the suggestion, we have changed in the revised version.

Lines: 115 (revised version), 425 (track changes manuscript).

**L113: ...protection, the most important being the Mount…**

Thanks for the suggestion, we have changed in the revised version.

Lines: 116 (revised version), 426 (track changes manuscript).

**L230: ...values, so continuous data need to be discretized.**

Thanks for the suggestion, we have changed the sentence in the revised version.

Lines: 256 (revised version), 1012 (track changes manuscript).

**L240: lose less information → minimize information loss**

Thanks for the suggestion, we have changed in the revised version.

Lines: 267 (revised version), 1091 (track changes manuscript).

**Table 5: Firec → Fires; ount of Day… → Count of Day…**

Thanks for the suggestion, we have changed in the revised version.

Lines: 269 (revised version), 1069 (track changes manuscript).

**L246: ...through a heuristic search**

Thanks for the suggestion, we have changed in the revised version.

Lines: 272 (revised version), 1096 (track changes manuscript).

**L255: ...66% of chance → ...66% chance**

Thanks for the suggestion, we have changed in the revised version.

Lines: 421 (revised version), 1765 (track changes manuscript).

**L266: linked → connection**

Thanks for the suggestion, we have changed the sentence in the revised version.

Lines: 307 (revised version), 1237 (track changes manuscript).

**L276: ...was overlaid with wildland areas.**

Thanks for the suggestion, we have changed in the revised version.

Lines: 328 (revised version), 1340 (track changes manuscript).

**L297: ...once, 34.8% twice, 23.1% has burned three times or more, and nearly 6% has been burnt more than 5 times.**

Thanks for the suggestion, we have changed in the revised version.

Lines: 356 (revised version), 1494 (track changes manuscript).

**L364: ...level of occurrence…**

Thanks for the suggestion, we have changed to "level of fire hazard", in order to keep the consistence in the revised version.

Lines: 420 (revised version), 1764 (track changes manuscript).

**Section 3.3 heading: Remove "and intermediate components" because it is unclear what these are.**

Thanks for the suggestion, we have removed it in the revised version.

Lines: 432 (revised version), 1865 (track changes manuscript).

**L374: vertical axis → column**

Thanks for the suggestion, we have changed in the revised version.

Lines: 438 (revised version), 1871 (track changes manuscript).

**L375: axis → column**

Thanks for the suggestion, we have changed in the revised version.

Lines: 439 (revised version), 1872 (track changes manuscript).

**L408: remote "Concerning data sources" and start the sentence with "Although"**

Thanks for the suggestion. We have changed the lines 408-419 as you suggested in the Minor Comments section as "*L408 to L419: This section of the discussion is confusing to understand and would benefit from a re- write*".

Lines: 474 (revised version), 2053 (track changes manuscript).

**L409: Unclear what "fruition" means.**

Thanks for the suggestion. We have changed the lines 408-419 as you suggested in the Minor Comments section as "*L408 to L419: This section of the discussion is confusing to understand and would benefit from a re- write*".

Lines: 475 (revised version), 2054 (track changes manuscript).

**L456: Capitalize "Traditional".**

Thanks for the suggestion, we have changed in the revised version.

Lines: 525 (revised version), 2259 (track changes manuscript).

**Table S3 caption: Area of ecosystem services potentially exposed to fire…**

Thanks for the suggestion, we have changed in the revised version.

S3, Table S3 has changed to S5, Table S3 in the revised supplementary materials.

---

## Author Response (AR2)

**Referee 2**

**The authors have adequately addressed my comments and concerns from the first review round. In particular, I appreciate the added clarity between fire hazard and fire risk presented in the manuscript. The research questions on page 4 are also a valuable addition. Finally, the clearer description of DAG is really helpful, and the inclusion of strength of influence is excellent. I have just a couple of very minor comments to be addressed below.**

**Minor comment:**

**In the response to reviewer document, there is a missing discussion of what was completed for one comment. Specifically:**

**"L55: The comment about "biodiversity is lost" is completely opposite to L22 where the introduction mentions "fires increase biodiversity". Please explain or correct this inconsistency. The positive or negative consequences of wildfires mainly depend on their size, intensity, and frequency. We will clarify the revised version as follows:"**

**- I did not find the same inconsistency when reading through the manuscript again. However, it would be great to get a summary of how this was addressed from the authors.**

Thank you very much for the suggestion. The sentence was not sufficiently clear, so we have added an additional sentence to enhance the understanding of how biodiversity or ecosystem services can be compromised through fire. As explained by various authors (Daily et al., 1997; Roces-Díaz et al., 2022; Tedim et al., 2020; Pausas et al., 2008; Regos et al., 2014; Castellnou et al., 2019), human intervention has significantly altered both ecosystems and fire regimes. In the case of ecosystems, two distinct phenomena have caused disruption: 1) The overprotection of ecosystems, coupled with the absence of herbivores, has resulted in an accumulation of fuel, particularly in Mediterranean ecosystems. Consequently, when a fire occurs, it burns with greater intensity due to the increased availability of combustible material. And 2) Overexploitation and degradation of certain areas have rendered some ecosystems highly fragile, making post-fire recovery more challenging.

Regarding fire regimes, the main issues are as follows: 1) The efficacy of fire suppression, leading to the phenomenon known as "fire trap" (Arno and Brown, 1991), which causes a shift in the natural fire regime. This alteration results in increased horizontal fire continuity (larger fire-prone areas) and further accumulation of fuel in forests (associated with ecosystem disruption and overprotection of forests). And 2) Human-caused ignitions, whether intentional or due to negligence, can affect highly degraded areas or areas with a substantial build-up of fire events.

The sentence has been modified in the revised manuscript as follows:

The consequences of wildfires exceed the loss of forest cover, vary over time and can be long-lasting. Some ecosystem properties and functions that deliver benefits to humans (Daily et al., 1997; Roces-Díaz et al., 2022), including biodiversity, may be lost. This diminish might happen when natural fire regimes and forest ecosystems are strongly altered by human intervention (Tedim et al., 2020; Arno and Brown, 1991), leading to an increase of fire extent, intensity and severity (Pausas et al., 2008; Regos et al., 2014; Castellnou et al., 2019).

References:

Arno, S. F. and Brown, J. K.: Overcoming the paradox in managing wildland fire, Western Wildlands, 17, 40–46, 1991.

Castellnou, M., Prat-Guitart, N., Arilla, E., Larrañaga, A., Nebot, E., Castellarnau, X., Vendrell, J., Pallàs, J., Herrera, J., Monturiol, M., Cespedes, J., Pagès, J., Gallardo, C., and Miralles, M.: Empowering strategic decision-making for wildfire management: avoiding the fear trap and creating a resilient landscape, fire ecol, 15, 31, https://doi.org/10.1186/s42408-019-0048-6, 2019.

Daily, G., Postel, S., Bawa, K., and Kaufman, L.: Nature's Services: Societal Dependence On Natural Ecosystems, Bibliovault OAI Repository, the University of Chicago Press, 1997.

Pausas, J. G., Llovet, J., Rodrigo, A., Vallejo, R., Pausas, J. G., Llovet, J., Rodrigo, A., and Vallejo, R.: Are wildfires a disaster in the Mediterranean basin? – A review, Int. J. Wildland Fire, 17, 713–723, https://doi.org/10.1071/WF07151, 2008.

Regos, A., Aquilué, N., Retana, J., Cáceres, M. D., and Brotons, L.: Using Unplanned Fires to Help Suppressing Future Large Fires in Mediterranean Forests, PLOS ONE, 9, e94906, https://doi.org/10.1371/journal.pone.0094906, 2014.

Roces-Díaz, J. V., Santín, C., Martínez-Vilalta, J., and Doerr, S. H.: A global synthesis of fire effects on ecosystem services of forests and woodlands, Frontiers in Ecology and the Environment, 20, 170–178, https://doi.org/10.1002/fee.2349, 2022.

Tedim, F., McCaffrey, S., Leone, V., Delogu, G. M., Castelnou, M., McGee, T. K., and Aranha, J.: 13 - What can we do differently about the extreme wildfire problem: An overview, in: Extreme Wildfire Events and Disasters, edited by: Tedim, F., Leone, V., and McGee, T. K., Elsevier, 233–263, https://doi.org/10.1016/B978-0-12-815721-3.00013-8, 2020.

**Technical corrections (line numbers are based on the tracked changes document):**

**L922: modeling, L924 modelling**

Thanks for the suggestion, we changed in the revised manuscript.

**L920: being 2012 → 2012 being**

Thanks for the suggestion, we changed in the revised manuscript.

**L1421: However, fires can not always be...**

Thanks for the suggestion, we changed in the revised manuscript.